# FAST AND SPACE-EFFICIENT FIXED-LENGTH PATH OPTIMIZATION

## ABSTRACT

Several optimization problems seek a *path* of *predetermined* length in a state space that minimizes a cost function. Conventionally, such problems are tackled by dynamic programming (DP) applying a Bellman-type equation. A prominent example is *Viterbi decoding*, which returns the path in a Hidden Markov Model that best explains a series of observations, with applications from bioinformatics to communication systems and speech recognition. However, DP-based solutions *(i)* exhaustively explore a search space linear in both state space size and path length in time quadratic in state space size, without exploiting data characteristics, and *(ii)* require memory commensurate with that search space to reconstruct the optimal path. In this paper, we propose Isabella (Di̲jk̲s̲tra-Be̲llma̲n), a novel framework that finds optimal paths of predetermined length in time- and space-efficient fashion by a combination of best-first-search, depth-first-search, and divide-and-conquer strategies. The *best-first-search* component avoids the exhaustive exploration of the search space using a priority queue; the *depth-first-search* component keeps the size of that queue in check; and the *divide-and-conquer* component constructs the optimal path recursively with low space complexity after determining its cost. We apply Isabella to Viterbi decoding, introducing algorithms that prioritize the most promising pathways and control memory consumption. To emphasize the generality of Isabella, we also instantiate it with an algorithm for histogram construction. To our knowledge, no previous work addresses such problems with this novel combination of strategies. Our experimental evaluation shows our solutions to be highly time- and space-efficient compared to standard dynamic programming.

## 1 INTRODUCTION

Several problems that call to find an optimal sequence of given length $L$ over a state space of size $n$ are conventionally solved by dynamic programming (DP). Two prominent examples are *Viterbi decoding*, i.e., finding a sequence of hidden states—a *Viterbi path*—in a Hidden Markov Model (HMM) that best explains an observed event sequence (Viterbi, 1967; 2006) and finding a sequence of bucket boundaries—a *V-optimal histogram*—over a value sequence that minimizes the aggregate Euclidean error when assigning one representative to each bucket (Bellman, 1961; Jagadish et al., 1998). As we show, the underlying formulation of the solution is the same.

Such DP algorithms use the solutions to *all* length-$k$ subproblems to solve *all* length-$k$+1 subproblems; to build the optimal sequence after reaching the target length $L$, one may follow one of two strategies: *(i)* the *memoization* strategy retains and backtracks over processed subproblem solutions per step, requiring $\Theta(nL)$ space and $\mathcal{O}(n^2L)$ time; *(ii)* the *in-place* strategy discards processed subproblem solutions and reruns the algorithm from scratch and on-demand backwards per step, hence requires $\mathcal{O}(n)$ space but $\mathcal{O}(n^2L^2)$ time. An algorithm recently proposed by Ciaperoni et al. (2022; 2024) constrains space to $\mathcal{O}(n)$ and runs in $\mathcal{O}(n^2L\log L)$ time, but still evaluates all subproblems exhaustively and *indiscriminately*, constraining its scalability to large problem instances. Still, in practice only a few subproblems aid the solution, while most are subpar building blocks; despite their long history, state-of-the-art DP algorithms fail to utilize this fact.

In this paper, we introduce Isabella (Di̲jk̲s̲tra-Be̲llma̲n), an algorithm design framework for the space- and time-efficient optimization of *predetermined*-length sequences, such as Viterbi paths and V-optimal histograms. Isabella combines best-first search, depth-first search, and divide-and-conquer

components. The best-first-search (BestFS) component prioritizes the most promising subproblems to avoid exhaustively exploring the subproblem space and enhance time efficiency. We also use bidirectional search and a bounding scheme for more effective subproblem prioritization. Of course the applicability of BestFS to find an optimal-cost path of *arbitrary* length is straigthforward (Russell & Norvig, 2010); however, its application to find an optimal-cost path of a *predetermined* length raises additional space requirements for *(i)* tabulating the optimal cost per state and step to enable backtracking of the optimal sequence and *(ii)* maintaining a correspondingly larger priority queue. To ameliorate these memory requirements, we amend BestFS with a depth-first search (DFS) strategy that prevents the priority queue from overextending and a divide-and-conquer (D&C) provision that omits tabulating all subproblem solutions. These innovations reduce both time and space requirements, while allowing one to control the ensuing tradeoff between time and memory demands via a tunable space budget parameter. To our best knowledge, no previous work tackles problems of this type as we do. We apply Isabella to Viterbi decoding (Viterbi, 1967) and V-optimal segmentation (Jagadish et al., 1998) and evaluate our solutions over real and synthetic data, showing that they gain in both runtime and memory vs. DP algorithms, especially under skewed distributions of path costs.

## 2 BACKGROUND AND RELATED WORK

**Dynamic programming** (DP) (Bellman, 1966) recursively decomposes a problem into subproblems, exploiting an *optimal substructure* property, by which a globally optimal solution combines locally optimal solutions. Nevertheless, its application is constrained by its computational requirements.

**Best-First search** (BestFS) (Pearl, 1984) repetitively expands a most *promising* partial solution to a problem. Dijkstra's algorithm (1959) finds a minimum-cost path of *arbitrary* length from a start to an end node; in each iteration it *explores*, i.e., visits the neighbors of, the nearest *unexplored* visited node, until it reaches the end node. This algorithm resembles a DP algorithm (Sniedovich, 2006), as it exploits optimal substructures to expand its solution. Thus, DP and BestFS are closely intertwined. We build upon this connection, applying BestFS to find an optimal-cost path of fixed length. We stress that, contrariwise to the Dijkstra algorithm, our task requires monitoring both path length and cost. $A^*$ (Foead et al., 2021) augments Dijkstra with a heuristic that prioritizes paths appearing closer to the end, while the search may proceed from both start and end by bidirectional search (Pohl, 1969).

**Hidden Markov Models** (HMMs) explain observation sequences. An HMM comprises a set of $K$ hidden states, each with probabilities to be an initial state, transition to other states, and emit an observation. *Decoding* seeks a sequence of states most likely to generate a sequence of observations:

**Problem 1** (Decoding). *Given an HMM and a sequence of $T$ observations $Y = \{y_1, y_2, \ldots, y_T\}$, find the sequence of hidden states $Q = \{s_1^*, s_2^*, \ldots, s_T^*\}$ that maximizes the likelihood $P(Q, Y)$.*

The Viterbi algorithm (Viterbi, 1967; 2006) solves Problem 1 optimally by DP; it finds application from networking and telecommunications (Viterbi, 2006) to speech recognition (Gales & Young, 2007; Braun et al., 2020), where it serves to find the most probable transcription for an input acoustic signal, or for forced alignment, the task of aligning orthographic transcriptions to audio recordings. In modern speech-recognition systems, the Viterbi algorithm runs on the composition of several small HMMs in which states represent words and their phonemes, to find the best transcription of a spoken utterance. However, this algorithm raises high memory and runtime requirements. A recent work (Jihyuk Jo, 2019) on HMM-based isolated word recognition employed a search heuristic, without proving its correctness. Another recent work (Ciaperoni et al., 2022) enhances the space efficiency of decoding at the cost of a runtime overhead. Other works reduce the state space representation for particular classes of HMMs (Siddiqi & Moore, 2005; Felzenszwalb et al., 2003). Still, the time complexity of the algorithm remains prohibitive for problem instances with large state space and long observation sequences. We aim to improve upon both the time and space efficiency of Viterbi decoding by provably correct best-first search, pruning, and bounding policies. This is a difficult undertaking, as indicated by derived lower bounds (Backurs & Tzamos, 2017).

**Histogram construction** calls to segment a data series to a predetermined number of buckets, each with one representative, to minimize the overall representation error:

**Problem 2** (Histogram Construction). *Given $I = \{x_1, \ldots x_n\}$, $x_i \in \mathbb{R}$, and $B \in \mathbb{Z}^+$, find a segmentation (or histogram) $H_B$ of $I$ into $B$ non-overlapping subsequences (or buckets) $I_b$ with associated bucket representatives $\hat{x}_b$ that minimizes error function $E_I(H_B)$.*

Problem 2 is central in data summarization (Halim et al., 2009). We focus on *V-optimal* histogram construction (Jagadish et al., 1998), i.e., Problem 2 with $E_I(H_B) = \sum_{b=1}^{B} E_b$, where $E_b = \sum_{x_i \in I_b} (x_i - \hat{x}_b)^2$, and $\hat{x}_b$ is the mean of values in bucket $I_b$. This extensively studied problem (Guha et al., 2006) is solved optimally by a DP (Jagadish et al., 1998) algorithm with quadratic dependence on $n$. Our framework offers significant gains in histogram construction and is extensible to any monotonic and distributive (Karras & Mamoulis, 2008) error measure $E_I(H_B)$.

**Semirings and Dioids** (Gondran & Minoux, 2008) A *semiring* is a 5-tuple $(D, \oplus, \otimes, \bar{0}, \bar{1})$, where $D$ is a non-empty set, $\oplus$ is a binary, associative, and commutative operator, $\otimes$ is a binary and associative operator, $\bar{0}$ is a neutral element for $\oplus$ (i.e., $x \oplus \bar{0} = x$, for all $x \in D$), $\bar{1}$ is a neutral element for $\otimes$ (i.e., $x \otimes \bar{1} = \bar{1} \otimes x = x$, for all $x \in D$), the operator $\otimes$ distributes over $\oplus$ and $\bar{0}$ is absorbing for $\otimes$ (i.e., $x \otimes \bar{0} = \bar{0} \otimes x = \bar{0}$, for all $x \in D$). A *selective dioid* is a semiring in which $\oplus$ is also selective (i.e., $(x \oplus y = x) \vee (x \oplus y = y)$, for all $x, y \in D$). Selective dioids provide an abstract expressive framework for shortest-path and DP problems (Mohri, 2002; Huang, 2008; Tziavelis et al., 2020).

## 3 THE ISABELLA FRAMEWORK

Isabella solves problems seeking a given-length sequence that optimizes a cost measure, which are typically solved by dynamic programming, visiting subproblems in a fixed order, oblivious of the fact that several subproblems do not contribute to the final solution. Contrariwise, Isabella solves subproblems in a best-first fashion until a stopping condition is met. Figure 1 shows an example.

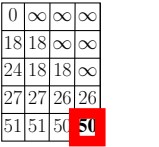

Figure 1: Each cell records the cost of a sub-problem; classic DP solves all sub-problems; Isabella finds the same solution, but avoids considering sub-problems in green.

Next, we define Isabella and apply it to Viterbi decoding (§ 3.2) and histogram construction (§ 3.4).

### 3.1 THE FRAMEWORK

Isabella starts out with the following components:

1. A *data space* $\mathcal{X}$ of $n$ elements endowed with a concept of *eligible sequence* $(x_j)$, $x_j \in \mathcal{X}$, $j \in \{1, \dots, \ell\}$, where $\ell$ is the length of $(x_j)$; a sequence is eligible if $i - 1 \in \mathcal{N}(i)$ for all $i \in \{2, \dots, \ell\}$ and a given *neighborhood* function $\mathcal{N}(\cdot)$.

2. The *set* $\mathbb{X}$ of all eligible sequences in $\mathcal{X}$.

3. A *gap function* $G(j, i)$ associating a value with the transition from item $j$ to item $i$.

4. A selective dioid $\mathcal{D} = (D, \oplus, \otimes, \bar{0}, \bar{1})$, which is used to express the *value function* for a sequence: $f(\mathbf{x}) = \bigotimes_{j=1}^{j=L-1} G(j, j+1)$.

5. A *problem* that seeks an eligible sequence $\mathbf{x}^* \in \mathbb{X}$ of *length* $L$ and *optimal* value $f(\mathbf{x}^*)$; sequences are compared via the $\oplus$ operator.

6. A recursive *function* $\mathsf{Opt}(i, \ell)$ that stores the *optimal* value for an eligible sequence of length $\ell$ ending at item $x_i \in \mathcal{X}$.

7. A *solution* to the problem in Item 5 by DP over sequences of increasing length from $\mathcal{X}$.

The solution in Item 7 finds an eligible sequence of $L$ data items $\mathbf{x}^* = \{x_1^*, x_2^*, \dots, x_L^*\}$ that optimizes $\mathsf{Opt}(\cdot, L)$; the selective dioid properties, in particular distributivity, guarantee correctness. The DP computation takes the form:

$$\mathsf{Opt}(i, \ell) = \bigoplus_{j \in \mathcal{N}(i)} \{\mathsf{Opt}(j, \ell - 1) \otimes G(j, i)\}. \tag{1}$$

The recursion of Equation 1 requires $\Theta(n^2 L)$ time and $\Theta(nL)$ space, iterating over items $i$ and lengths $\ell$ and storing, for each $(i, \ell)$ pair, a predecessor needed to backtrack the optimal sequence.

The entailed solution may be implemented by either DP or *token passing* (Young et al., 1989); both calculate *all* solutions of length $\ell$, $\mathsf{Opt}(\cdot, \ell)$, before those of length $\ell+1$, $\mathsf{Opt}(\cdot, \ell+1)$, by *breadth-first search*. DP draws from solutions of length $\ell$ to build each solution of length $\ell + 1$ by a *pull* approach. Token passing broadcasts a *token* for each solution of length $\ell$ to its continuations of length $\ell + 1$ by a *push* approach. In both cases, solutions solidify at length $\ell$ before moving to length $\ell + 1$.

Isabella abolishes this *breadth-first* orientation in favor of a *best-first* one; it organizes sub-problem solutions (represented by tokens) in a priority queue $\mathbf{Q}$ in which it initially inserts all tokens $(\cdot, 1)$. Thereafter, in each step, it selects the most promising token $(j, \ell)$ from $\mathbf{Q}$ and, for each eligible successor $(i, \ell + 1)$ not already extracted from $\mathbf{Q}$, it computes $\mathsf{Opt}(j, \ell) + G(j, i)$ and updates the priority of $(i, \ell + 1)$ in $\mathbf{Q}$ accordingly. Isabella resembles Dijkstra's shortest-path algorithm (1959); however, whereas Dijkstra minimizes a cost objective regardless of length (i.e., number of steps), Isabella optimizes the objective under a *fixed*-length constraint. In the worst case, it examines all sequence continuations for each length, hence takes $\mathcal{O}(nL(n + \log nL))$ time, where the $\log nL$ term expresses the overhead of maintaining the priority queue, more compactly $\mathcal{O}(nL(n + \log L))$. However, in practice, it gains performance as it quickly derives tokens corresponding to DP table cells without considering all possible paths and does not produce some tokens at all. As we will see in Section 4, this pruning capacity results in significant runtime savings, particularly in real problem instances. In the best case, Isabella may produce only $L$ tokens, each in constant time.

Nevertheless, the priority queue $\mathbf{Q}$ may reach size $\Theta(nL)$, holding one token for each state-length pair, while we need to keep all tokens in memory after popping them from $\mathbf{Q}$ to facilitate backtracking the solution sequence. To contain this space demand, we eschew backtracking and instead reconstruct the solution by a *divide-and-conquer* strategy as in (Ciaperoni et al., 2022; 2024) that achieves $\mathcal{O}(n)$ space and $\mathcal{O}(n^2 L \log L)$ time; we keep the size of $\mathbf{Q}$ in $\mathcal{O}(n)$ by a *depth-first-search* strategy.

In our Isabella variants, we anticipate, by bounding, the values a sequence may achieve as it expands and prioritize tokens according to those anticipatory values. We also derive bidirectional-search variants that produce both prefixes and suffixes of sequences until they reach the target length.

## 3.2 THE MINT ALGORITHM

The Viterbi algorithm selects a sequence of $T$ states $Q = \{s_1^*, s_2^*, \ldots, s_T^*\}$ from a universe of $K$ HMM states $S = \{s_1, s_2, \ldots, s_K\}$ that is *most likely* to have generated a sequence of $T$ observations $Y = \{y_1, y_2, \ldots, y_T\}$. $Q$ is called *Viterbi path*. By the Markov property, the likelihood to be in a state depends only on the previous state. Therefore, the Viterbi algorithm uses the DP recursion:

$$\mathbf{T}[s_i, 1] = \pi_{s_i} \cdot B_{s_i, y_1},$$
$$\mathbf{T}[s_i, t] = \max_{s_h \in \mathcal{N}_{in}(s_i)} \{\mathbf{T}[s_h, t-1] \cdot A_{s_h, s_i}\} \cdot B_{s_i, y_t} \tag{2}$$

where $\mathbf{T}[s_i, t]$ stores the probability of the most likely path ending at state $s_i$ in $t$ steps, or *time frames*, $\mathcal{N}_{in}(s_i)$ is the set of in-neighbors of $s_i$, $\pi_i$ is the initial probability of $s_i$, $A_{s_h, s_i}$ is the probability of transiting from state $s_h$ to state $s_i$ on a directed graph $G$ capturing eligible transitions in the HMM, and $B_{s_i, y_t}$ is the probability of observing $y_t$ at state $s_i$. This setting suits Isabella as follows:

1. The *data space* $\mathcal{X}$ is the universe of $K$ hidden states $S = \{s_1, s_2, \ldots, s_K\}$ and an *eligible sequence* $(x_j)$, $x_j \in \mathcal{X}$, $j \in \{1, \ldots, \ell\}$ is a path of consecutive states in the HMM graph $G$.
2. The *set* $\mathbb{X}$ of all eligible sequences in $\mathcal{X}$ is the set of all possible paths in the given HMM.
3. The *gap function* $G(j, i)$ is expressed as $A_{s_j, s_i} B_{s_i y_i}$, or, in the domain of log-probabilities, as $\log A_{s_j, s_i} + \log B_{s_i y_i}$.
4. The *selective dioid* is $([0, 1], \max, \cdot, 0, 1)$, or, in the domain of log-probabilities, $([-\infty, 0], \max, +, -\infty, 0)$. Thus, the *value function* $f$ assigns probabilities to paths, given the sequence of observations $Y = \{y_1, y_2, \ldots, y_T\}$; the probability that $Y$ is generated by a sequence of hidden states $Q = \{s_1, s_2, \ldots, s_T\}$ is $P(Q, Y) = \pi_{s_1} \cdot B_{s_1 y_1} \prod_{i=2}^{T} A_{s_{i-1} s_i} \cdot B_{s_i y_i}$, where $\pi(s_1)$, $A_{s_{i-1} s_i}$, and $B_{s_i y_i}$ are defined as above.
5. The *problem* seeks an eligible sequence of states $Q$ of length $T$ that best explains the given sequence of observations $Y$, i.e., maximizes probability ($\oplus = \max$).
6. The recursive *function* $\mathsf{Opt}(i, \ell)$ that stores the *optimal* value for an eligible sequence of length $\ell$ that ends at data item $x_i \in \mathcal{X}$ is the function $\mathbf{T}[s_i, t = \ell]$.
7. The *solution* by DP over sequences of increasing length from $\mathcal{X}$ is given by Equation 2.

The recursion of Equation 2 requires $\mathcal{O}(K^2 T)$ time and $\mathcal{O}(KT)$ space, as it iterates over states $s_i$ and time frames $t$. For the sake of efficiency and accuracy, we replace products of likelihoods by sums of log-likelihoods. In case the structure of $G$ is known, we iterate only over states $s_h$ that link to state $s_i$, hence visit each HMM graph edge only once; then time complexity becomes $\mathcal{O}((K + |E|)T)$.

The Viterbi algorithm and its *token passing* variant (Young et al., 1989) operate by *breadth-first search*. MINT replaces this strategy with *best-first* search. It organizes sub-problem solutions in a priority queue $\mathbf{Q}$ and, in each step, expands the most promising one. It resembles Dijkstra's algorithm (1959), yet Dijkstra finds a shortest path without a constraint, while a Viterbi path optimizes path likelihood under a *fixed* number of steps. MINT also resembles the Viterbi algorithm in deriving $\mathbf{T}[s_i, t + 1]$ entries from $\mathbf{T}[s_h, t]$ entries. Still, while Viterbi calculates *all* $\mathbf{T}[\cdot, t]$ entries before $\mathbf{T}[\cdot, t + 1]$ entries, MINT first inserts all tokens $(s_h, 1)$ in $\mathbf{Q}$ and thereafter picks the most promising token $(s_h, t)$ from $\mathbf{Q}$ and, for each *outgoing* neighbor $s_i \in \mathcal{N}_{out}(s_h)$ such that $(s_i, t + 1)$ has not been extracted from $\mathbf{Q}$, it computes $\mathbf{T}[s_h, t] \cdot A_{s_h, s_i} \cdot B_{s_i, y_{t+1}}$ and updates the priority of $(s_i, t + 1)$ accordingly. In effect, MINT computes $\mathbf{T}[s_i, t + 1]$ values by *push* operations over outgoing neighbors $s_i$ of each $s_h$ popped from $\mathbf{Q}$. In the worst case, it visits each HMM edge once for each time frame, hence takes $\mathcal{O}((K \log KT + |E|)T)$ time, where $\log KT$ expresses the queue overhead. In practicality, it never considers some HMM edges and DP cells. In the following, we illustrate four MINT variants.

---

**Algorithm 1:** Standard MINT

**Data:** HMM graph $G$, transition and emission probabilities $A$ and $B$, observations $Y$, and initial state $s$.
**Result:** Viterbi Path Log-Likelihood $\max_{s_i} \mathbf{T}[s_i, T]$.
1  $\mathbf{Q} \leftarrow Queue((s, 1), p(s, 1) = -\log B_{s, y_1})$;
2  $\mathbf{V} \leftarrow \{\}$;
3  **while** $\mathbf{Q} \neq \emptyset$ **do**
4     $(s_i, t), p_i \leftarrow \mathbf{Q}.pop()$; // (state, frame), priority
5     **if** $t = T$ **then break**;
6     $\mathbf{V}.add((s_i, t))$;
7     **for** $s_j$ *in* $G[s_i]$ **do**
8        **if** $(s_j, t + 1) \notin \mathbf{V}$ **then**
9           $d \leftarrow p_i - \log A_{s_i, s_j} - \log B_{s_j, y_{t+1}}$;
10          **if** $(s_j, t + 1) \notin \mathbf{Q}$ **then** $\mathbf{Q}.insert((s_j, t + 1), p(s_j, t + 1) = d)$;
11          **if** $\mathbf{Q}[(s_j, t + 1)] > d$ **then** $\mathbf{Q}.update((s_j, t + 1), p(s_j, t + 1) = d)$;
12 **return** $-p_i$;

---

**Standard MINT.** To achieve the $\max_{s_i} \mathbf{T}[s_i, T]$ objective, we equivalently *minimize* the positive path log-likelihood, $-\log P(Q, Y) \geq 0$, or *cost* of a path. Standard MINT uses a priority queue $\mathbf{Q}$ in which tokens $(s_i, t)$ are inserted and updated in logarithmic time and looked up in constant time. For a given path $Q$ ending at state $i$ at frame $t$, we define its priority as $p(s_i, t) = -\log P(Q, Y)$, with $Q = \{s^1, s^2, \ldots, s^t\}$, such that $s^t = s_i$ and $Y = \{y_1, y_2, \ldots, y_t\}$. Algorithm 1 gives the pseudocode. First we insert in $\mathbf{Q}$ a token for each state in the first frame with priority $-\pi_{s_i} - \log B_{s_i, y_1}$. If a start state $s$ is given, we only enqueue a token for $s$ at $t = 0$. The main loop iterates until reaching the last frame $t = T$. In each iteration, we dequeue from $\mathbf{Q}$ the top token $(s_i, t)$, add it to a set $\mathbf{V}$ of visited tokens, and compute the cost of reaching $(s_j, t + 1)$ via $(s_i, t)$ for each out-neighbor $s_j$ of $s_i$ that has no such token in $\mathbf{V}$ and insert or update $(s_j, t + 1)$ in $\mathbf{Q}$ by that cost. When a pair $(s, T)$ is dequeued, no path spanning $T$ frames at lower cost exists, hence we may return its cost as the Viterbi path log-likelihood $\max_{s_i} \mathbf{T}[s_i, T]$. All proofs are in Appendix A.

**Proposition 1.** *Standard MINT is correct.*

To return the optimal path, MINT by defaults appends a path to each token and, when updating the priority of $(s_j, t + 1)$, also updates the corresponding path to $s_j$. We also craft a variant, MINT-Backtracking, that stores only the *predecessor* of each token, retains all explored tokens, and at the end constructs the optimal path by *backtracking*. Next, we present three extensions to MINT.

**MINT Bound.** We propose a variant of MINT that orders tokens by lower-bounding the path cost from each token $(s_i, t)$ until the final frame $T$ using a *lower bound* $\hat{c}_1$ on the cost for moving from one frame to another for the remaining $T - t$ frames. We call the ensuing algorithm MINT Bound.

We first insert all states (or the source state, if given) in $\mathbf{Q}$ with $p(s, 1) = T \cdot \hat{c}_1$. As the search proceeds, we replace lower bounds with exact costs. The priority of $(s_i, t)$ is $p(s_i, t) = -\log P(Q, Y) + (T - t) \cdot \hat{c}_1$, $Q$ being the current optimal path ending at $s_i$ in $t$ frames; we compute the priority of a neighbour $s_j$ for insertion or update in $\mathbf{Q}$ as $p(s_i, t) - \hat{c}_1 - \log A_{s_i, s_j} - \log B_{s_j, y_{t+1}}$. Upon reaching the last frame,

all lower bounds capture exact costs. Setting $\hat{c}_1$ as the lowest value of $-\log A_{s_i,s_j} - \log B_{s_j,y_{t+1}}$ over all edges $(s_i, s_j)$ at any $t$, found by pre-processing, ensures correctness.

**Proposition 2.** *MINT Bound is correct.*

MINT Bound encases information from unexamined time frames in the BestFS criterion and further explores already well-explored paths without compromising the correctness of Standard MINT.

**Bidirectional MINT** strives for an even more efficiency by bi-directional search (Pohl, 1969) to explore solutions both forward from the start and backward from the end time frame, using edges in the reverse direction. We denote the graph in which all edges are reversed in direction as $G_{rev}$.

---

**Algorithm 2:** Bidirectional MINT

**Data:** HMM graph $G$, transition and emission probabilities $A$ and $B$, observations $Y$, and initial and final states $source$ and $target$.
**Result:** Viterbi Path Log-Likelihood $\max_{s_i} \mathbf{T}[s_i, T]$.

1   $\mathbf{Q}_f \leftarrow Queue((source, 1), p(source, 1) = -\log B_{source, y_1})$;
2   $\mathbf{Q}_b \leftarrow Queue((target, T), p(target, T) = 0)$;
3   $\mathbf{V}_f \leftarrow \{\}; \mathbf{V}_b \leftarrow \{\}; \mu \leftarrow \infty$;
4   **while** $\mathbf{Q}_b \neq \emptyset \wedge \mathbf{Q}_f \neq \emptyset$ **do**
5      $(s_i^f, t^f), p_i^f \leftarrow \mathbf{Q}_f.\text{pop}(); (s_i^b, t^b), p_i^b \leftarrow \mathbf{Q}_b.\text{pop}()$;
6      $d_f[s_i^f] \leftarrow p_i^f; d_b[s_i^b] \leftarrow p_i^b$;
7      $\mathbf{V}_f.\text{add}((s_i^f, t^f)); \mathbf{V}_b.\text{add}((s_i^b, t^b))$;
8      **if** $t^f < T$ **then**
9         **for** $s_j$ *in* $G[s_i^f]$ **do**
10            Update $\mathbf{Q}_f$ for $(s_j, t+1)$;
11            **if** $(s_j, t+1) \in \mathbf{V}_b \wedge d_f[(s_i^f, t^f)] - \log A_{s_i^f, s_j} - \log B_{s_j, y_{t+1}} + d_b[(s_j, t+1)] < \mu$ **then**
12               $\mu = d_f[(s_i^f, t^f)] - \log A_{s_i^f, s_j} - \log B_{s_j, y_{t+1}} + d_b[(s_j, t+1)]$;
13      **if** $t^b > 1$ **then**
14         **for** $s_j$ *in* $G_{rev}[s_i^b]$ **do**
15            Update $\mathbf{Q}_b$ for $(s_j, t-1)$;
16            **if** $(s_j, t-1) \in \mathbf{V}_f \wedge d_b[(s_i^b, t^b)] - \log A_{s_j, s_i^b} - \log B_{s_i^b, y_t} + d_f[(s_j, t-1)] < \mu$ **then**
17               $\mu = d_b[(s_i^b, t^b)] - \log A_{s_j, s_i^b} - \log B_{s_i^b, y_t} + d_f[(s_j, t-1)]$;
18      **if** $d_f[(s_i^f, t^f)] + d_b[(s_i^b, t^b)] \geq \mu$ **then break**;
19   **return** $-\mu$;

---

In each direction, the search proceeds as in Standard MINT, yet with two priority queues, $\mathbf{Q}_f$ for forward search and $\mathbf{Q}_b$ for backward search. $\mathbf{Q}_f$ is initialized as in Standard MINT. Similarly, we start by enqueueing all states in $\mathbf{Q}_b$, or only the final state, if given. If an initial state is given and a final state is not given, we find, in pre-processing, all states $\mathcal{S}(T)$ reachable from the initial state in $T$ frames, and initiate $\mathbf{Q}_b$ with those. Algorithm 2 shows the pseudocode, which assumes that both an initial and final state are given. In each iteration, we expand both[1] searches, handling queues as in Standard MINT. Upon extracting $(s_i^f, t^f)$ from $\mathbf{Q}_f$ and $(s_i^b, t^b)$ from $\mathbf{Q}_b$, we update the associated visited-token sets, $\mathbf{V}_f$ and $\mathbf{V}_b$, and store associated costs in arrays $d_f$ and $d_b$. We then consider tokens $(s_j^f, t^f + 1)$ for all neighbors $s_j^f$ of $s_i^f$ in $G$ and tokens $(s_j^b, t^b - 1)$ for all neighbors $s_j^b$ of $s_i^b$ in the graph $G_{rev}$. In Lines 10 and 15 we omit the details, which are found in Algorithm 1, Lines 25–14.

When the two sides meet generating a path of length $T$, we update the hitherto best path cost $\mu$, if the newly found path improves on it. Such a path is provably optimal, hence the algorithm terminates, when the sum of costs of tokens dequeued from $\mathbf{Q}_f$ and $\mathbf{Q}_f$ exceeds $\mu$. To avoid double-counting emission probabilities, we add the emission probability of the last vertex visited while building a path only in the priority of $\mathbf{Q}_f$, thus the cost of any path through $(w, t)$ is $d_f[(w, t)] + d_b[(w, t)]$.

**Proposition 3.** *Bidirectional MINT is correct.*

**Bidirectional MINT Bound** combines Bidirectional MINT and MINT Bound in one algorithm that searches in both directions and lower-bounds the total $T$-frames-long path costs used as priority values in both queues. The search in both directions follows the order determined by the cost of arriving to a state in a given number of steps (frames) plus a lower bound on the cost of arriving from there to the the end of the path. The single-frame lower bound for the forward search is as in MINT Bound. For backward search, it is the lowest cost of moving from a frame to the next one in the reverse HMM graph $G_{rev}$, i.e., the lowest value of $-\log A_{s_j, s_i} - \log B_{s_j, y_t}$ over all edges $(s_j, s_i)$

---

[1] A more refined strategy would choose which side to expand, representing an opportunity for future work.

in $G_{rev}$. We obtain both bounds by pre-processing with a single graph traversal. The correctness of Bidirectional MINT Bound follows from the correctness of MINT Bound and Bidirectional MINT.

### 3.3 LINEAR-SPACE MINT (MINT-LS)

The backtracking discussed in Section 3.2 keeps all tokens in memory even after they are popped from $\mathbf{Q}$, incurring a $\mathcal{O}(KT)$ space complexity. Instead, we reconstruct the solution by a *divide-and-conquer* strategy and keep the size of $\mathbf{Q}$ in $\mathcal{O}(K)$ by a *depth-first-search* strategy.

To avoid materializing paths, we adapt the logic of SIEVE-Middlepath (Ciaperoni et al., 2022) to MINT. SIEVE-Middlepath reformulates the Viterbi algorithm by a divide-and-conquer strategy. Instead of tabulating all subproblem solutions, it only maintains those for the most recent frame, which it uses to solve subproblems in the next frame; it records, with each solution, the edge at the *middle* frame of the solution path (the *middle pair*), which it identifies upon reaching that middle frame. After termination, it recursively reruns on the $L/2$-hop predecessors and successors of the solution middle pair to reconstruct the full path. Likewise, MINT-LS identifies a middle pair upon reaching a middle frame and stores it in the solution token. After establishing the best token at the last frame, it reruns among the predecessors and successors of that token middle pair to reconstruct the entire Viterbi path. Middle pairs are retrieved in orderly fashion, as in an in-order tree traversal (Ciaperoni et al., 2022). Eschewing path materialization requires $\mathcal{O}(K^2 T \log T)$ time, yet does not guarantee $\mathcal{O}(K)$ space, as the size of $\mathbf{Q}$ may still grow to $\Theta(KT)$. Next, we address the queue size.

We keep the size of $\mathbf{Q}$ in check via a depth-first-search (DFS) strategy which is novel in this context. When $\mathbf{Q}$ exceeds a predefined size threshold $\theta$ after we insert a token $(s_j, t)$, we pick the lowest-cost token among the set of tokens for $s_j$, $\mathcal{S}^j = \{(s_j, \cdot)\}$, say $(s_j, t^*)$, and produce all its derivatives on demand via a DFS traversal of the HMM graph $G$ from $s_j$. Each branch of DFS terminates when it either reaches the last frame or injects a token into the token set of a state $s_{j'}$ that substitutes a pre-existing token without memory increase. The paths DFS explores identify and pass on middle pairs as usual. By virtue of this DFS operation, we maintain a priority queue of size $\mathcal{O}(K)$. In practice, we let the priority queue size threshold $\theta$ grow slightly with $T$, as an increase of $T$ without a commensurate alteration of $\theta$ triggers more DFS calls and hence burdens runtime.

Algorithm 4 in Appendix E provides the pseudocode of MINT-LS, which invokes the DFS subroutine of Algorithm 5 therein. MINT-LS propagates tokens like MINT does, yet it stores in each token $(s_i, t)$ its predecessor state and the currently valid middle pair for the path this token belongs to. These details help identify the subproblems to be solved recursively. Upon reaching the final frame (and final state, if given), it extracts the middle pair associated with the solution path and reruns recursively among $N_p$-hop predecessors of the middle pair in the $N_p$ frames preceding it and among $N_s$-hop successors of the middle pair in the $N_s$ frames following it, found by breadth-first search (Lines 15 and 22). Besides, MINT-LS eschews storing new tokens that would enlarge the queue beyond a threshold $\theta$ by invoking the DFS subroutine that only stores tokens replacing others, or at the last frame.

### 3.4 THE TECH ALGORITHM

We apply Isabella to another optimal-sequence problem, V-Optimal histogram construction (Jagadish et al., 1998), outlined in Section 2. A V-optimal (V for *variance*) histogram uses the mean value $\hat{x} = \frac{1}{i-j+1} \sum_{k=j}^{i} x_k$ as a representative to minimize the *Euclidean* error in a bucket $I_b$ extending from the $j^{\text{th}}$ to the $i^{\text{th}}$ value in the sequence, $E(j, i) = \sum_{k=j}^{i} (x_k - \hat{x})^2$. The total approximation error is aggregated over all buckets. V-OPT histogram construction algorithms use incremental sums and sum of squares, stored in arrays $S$ and $SS$, respectively, to obtain any $E(i, j)$ efficiently as:

$$E(j, i) = (SS[i] - SS[j-1]) - \frac{(S[i] - S[j-1])^2}{(i - j + 1)}. \tag{3}$$

The algorithm finds, for each combination $(i, b)$ of a value index and number of buckets, the cost of the optimal $b$-bucket histogram covering the first $i$ values in the sequence, as:

$$E^*(i, b) = \min_{1 \le j < i} E^*(j, b-1) + E(j+1, i), \tag{4}$$

Note that $E^*(i, b) = 0$ for $i \le b$ and $E^*(i, 1) = E(1, i)$. The algorithm returns the optimal cost $E^*(n, B)$ of the V-optimal histogram for a sequence of length $n$ and $B$ buckets and constructs the histogram by backtracking. We map the histogram construction problem to Isabella as follows:

1. The *data space* $\mathcal{X}$ is the data series $I = \{x_1, \ldots x_n\}$ and an *eligible sequence* $(x_j)$, $x_j \in \mathcal{X}$, $j \in \{1, \ldots, \ell\}$ comprises *ordered* items $(x_{i_1}, x_{i_2}, \ldots, x_{i_k})$ denoting segment boundaries.
2. The *set* $\mathbb{X}$ of eligible sequences in $\mathcal{X}$ includes all ordered sequences ending at $x_n$.
3. The *gap function* $G(j, i)$ is the bucket error $E(j, i)$, as above.
4. The *selective dioid* is $(\mathbb{R} \cup \{+\infty\}, \min, +, +\infty, 0)$. Thus, the *value function* $f : \mathbb{X} \to \mathbb{R}$ assigns an approximation error $\sum_{b=1}^{B} E_b$ to a histogram.
5. The *problem* seeks an error-minimizing eligible histogram of $B$ boundaries ($\oplus = \min$).
6. The recursive *function* $\mathrm{Opt}(i, \ell)$ that stores the *optimal* value for an eligible sequence of length $\ell$ ending at item $x_i \in \mathcal{X}$ is the function $E^*(i, b = \ell)$.
7. The *solution* by DP over sequences of increasing length from $\mathcal{X}$ is given by Equation 4.

The DP algorithm for V-OPT histogram construction (Jagadish et al., 1998) by Equation 4 requires $O(n^2 B)$ time and $O(nB)$ space. We discuss four variants of histogram construction by Isabella, TECH (Time-Efficient Histogram), corresponding to the four variants of MINT.

---

**Algorithm 3:** Standard TECH

**Data:** Input sequence $I$, integer $B$.
**Result:** V-Optimal Histogram Error $E^*(n, B)$.
1   $\mathbf{Q} \leftarrow Queue((1, 1), priority = 0)$;
2   $S \leftarrow []; SS \leftarrow []; \mathbf{V} \leftarrow \{\}; n \leftarrow I.length$;
3   $S[1] \leftarrow I[1]; SS[1] \leftarrow I[1]^2$;
4   **for** $i \in \{2, \ldots, n\}$ **do**
5      $S[i] \leftarrow S[i-1] + I[i]; SS[i] \leftarrow SS[i-1] + I[i]^2$;
6   **for** $j \in \{2, \ldots, n - (B-1)\}$ **do**
7      $\mathbf{Q}.insert((j, 1), priority = E(1, j))$;
8   **while** $\mathbf{Q} \neq \emptyset$ **do**
9      $(i, b), p \leftarrow \mathbf{Q}.pop()$; // (values,buckets),priority
10      $\mathbf{V}.add((i, b))$;
11      **if** $b = B \wedge i = n$ **then** break;
12      **if** $b < B$ **then**
13         **for** $j \in \{i+1, \ldots, n - B + b + 1\}$ **do**
14            **if** $(j, b+1) \notin \mathbf{V}$ **then**
15               $d \leftarrow p + E(i, j)$;
16               **if** $(j, b+1) \notin \mathbf{Q}$ **then**
17                  $\mathbf{Q}.insert((j, b+1), priority = p + E(i+1, j))$
18               **if** $\mathbf{Q}[(j, b+1)] \geq p + E(i+1, j)$ **then**
19                  $\mathbf{Q}.update((j, b+1), priority = p + E(i+1, j))$
20   **return** $p$;

---

**Standard TECH.** Algorithm 3 presents Standard TECH, which, like MINT, employs a priority queue $\mathbf{Q}$ to prune computations, where the priority of entry $(i, b)$ is the cost of the $b$-bucket histogram for the first $i$ values, $p(i, b) = E^*(i, b)$. After computing the $S$ and $SS$ arrays, used to compute the error measure by Equation 3, in each iteration, TECH dequeues the pair $(i, b)$ of lowest error, adds it to a set $\mathbf{V}$ of visited tokens, and, provided $b < B$, computes via $(i, b)$ the error for each pair $(j, b+1)$ with $j > i$ that is not in $\mathbf{V}$ and inserts or updates $(j, b+1)$ in $\mathbf{Q}$ accordingly; thereby, it explores possible next buckets. We do not iterate over $(j, b+1)$ pairs for all $j > i$, but stop at $j = n - B + b + 1$ since there must be at least $B - b - 1$ values after $j$ to make $B$ buckets in total. The algorithm terminates after it dequeues $(n, B)$. Correctness follows as in Proposition 1: after $(n, B)$ is dequeued, there can be no lower-cost histogram of the same series and $B$. For further pruning, we use an upper bound $UB[i, b]$ on the cost of a $(B - b)$-bucket histogram for the series $\{i + 1, \ldots n\}$, derived in Proposition 4 below. If, after visiting $(i, b)$, we find $j^* > i$ such that $E(i, j^*) \geq UB[i, b]$, we eschew computing $E(i, j)$ for $j \geq j^*$, as we have already exceeded the upper bound on the error therefrom.

**TECH Bound.** This variant uses bounds on the cost of a $B$-bucket histogram instead of the cost of a partial histogram with $b \leq B$ buckets. Given $(i, b)$, we partition the series $\{i + 1, \ldots n\}$ in $B - b$ equal-width buckets. The minimum error among such buckets is a lower bound to the V-optimal histogram cost, while the sum of those errors is an upper bound, which we may use for pruning.

**Proposition 4.** *The minimum error* $\min_b E_b$ *among $b$ buckets of an equal-width partitioning a sequence $I$ is a lower bound to the V-optimal histogram cost and* $\sum_b E_b$ *is an upper bound.*

TECH Bound adds to the priority of each pair $(i, b)$ the associated lower bound $LB[i, b]$. Upon arrival at the end of the series, there is no lower bound to be added, and the algorithm outputs the same cost as Standard TECH. We find these bounds by building equal-width histograms in a single pre-processing step. The correctness of TECH Bound follows as the correctness of MINT Bound.

Other TECH variants work by analogy to MINT variants. **Bidirectional TECH** applies forward and backward BestFS in alternation. When we pop a pair $(i, b)$ from $\mathbf{Q}_f$, we consider entries $(j, b+1)$ with $j > i$, and, if the backward search has already visited $(j, B - b - 1)$, we check whether the cost $d_f[(i, b)] + E(i, j) + d_b[(j, B - b - 1)]$ improves upon the current best cost $\mu$, and likewise in backward search. The search terminates when the sum of the costs of pairs from both queues exceeds the current best cost $\mu$. **Bidirectional TECH Bound** combines Bidirectional TECH with TECH Bound, with reversed lower bounds that consider sequences of the form $\{1, \dots i\}$ rather than $\{i+1, \dots n\}$. **TECH-LS** and variants thereof limit space needs; in place of a *middle pair* of states, they detect a *middle bucket* that splits the data series in halves. Previous work applied such a space-saving solution with dynamic programming (Guha, 2005).

## 4 EXPERIMENTS

**Implementation.** We ran experiments on a machine with a $2 \times 12$ core Xeon E5 2680 v3 2.50 GHz processor and 128 GB memory. Our implementation and experimental data are available[2] online.

**Baselines.** On Viterbi decoding, we use the Viterbi algorithm (§3.2) with edge-aware implementation as a baseline. We also juxtapose MINT-LS to a Viterbi variant that naïvely recomputes sub-paths for space-efficiency (Ciaperoni et al., 2022), checkpoint Viterbi (Tarnas & Hughey, 1998), which achieves limited space efficiency by segmenting the sequence into $\sqrt{T}$ parts, and SIEVE variants (Ciaperoni et al., 2022), including SIEVE-Middlepath (from which MINT-LS draws), Standard SIEVE (which recursively partitions the state space rather than the sequence), and SIEVE-Hyperloglog, which uses approximate counts of the number of predecessors and successors of a state (Flajolet et al., 2007). On histogram construction, we use as a baseline the DP V-optimal histogram construction algorithm (Jagadish et al., 1998) (§ 3.4). Appendix B describes data, measures, and parameters.

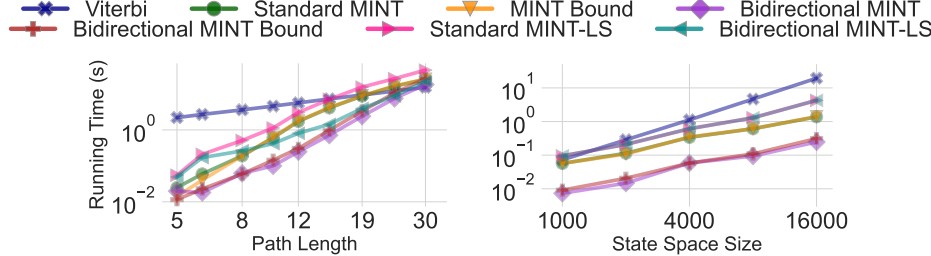

Figure 2: Decoding, synthetic data. Runtime vs. path length and state space size; axes in log scale.

### 4.1 RESULTS ON DECODING

**Synthetic data, runtime vs. $T$ and $K$.** Figure 2 plots runtime as a function of path length $T$ and state space size $K$ on Erdős–Rényi data. MINT accelerates decoding significantly, thanks to visiting only a small fraction of the tokens visited by Viterbi. Futher, MINT-LS exhibits only a marginal runtime overhead compared to MINT. Still, the time to visit a single token is lower in Viterbi as MINT incurs more overhead for managing the priority queue. Therefore, as the figure illustrates, savings drop as $T$ grows. This result arises from the data generating model, which reflects a worst-case scenario whereby the likelihoods of all paths converge to similar values for large enough $T$, therefore MINT visits too many tokens. As we will see, when the probability is more concentrated over a limited subset of paths, as in real-world speech recognition data, MINT yields higher savings; it identifies the most promising paths and neglects the others, thus visiting only a small fraction of the tokens visited by the Viterbi algorithm. Appendix C presents further results on runtime and other aspects.

**Real data, runtime vs. $T$ and $K$.** Figure 3 plots our results on forced-alignment and standard decoding with real data. Here, MINT results in savings of up to three orders of magnitude compared to Viterbi, as on real data it only explores a few promising paths. More remarkably, enlarging the state space by taking larger snowball samples of the HMM does not impact MINT, as it already avoids non-sampled states. Yet, state space growth greatly affects Viterbi. Moreover, MINT-LS variants gain up to three orders of magnitude lower runtime than Viterbi even while controlling memory too.

---

[2]https://github.com/TimeEfficientDP/BestFirst

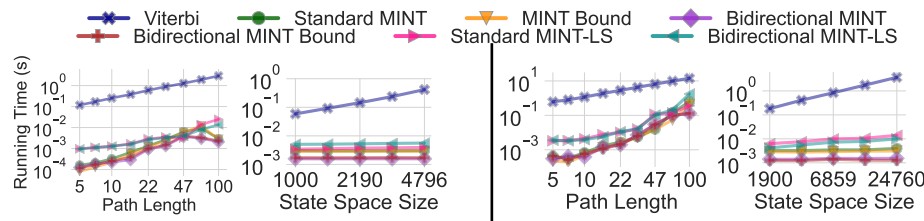

Figure 3: Decoding, real data; forced alignment (L), standard decoding (R); runtime vs. $T$ and $K$.

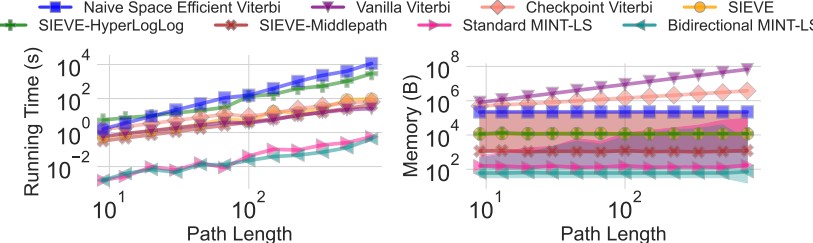

Figure 4: Decoding, real forced alignment data; runtime (left) and memory consumption (right) vs $T$. Shaded regions indicate the range of memory consumed over recursive calls.

**Real data, runtime and memory vs. $T$.** Figure 4 portrays runtime and memory needs, including those of DP-based SIEVE (Ciaperoni et al., 2022) variants, and plots the minimum, maximum and median memory usage across recursion levels. While DP-based baselines require *static* memory at each level, MINT-LS's requirement is *dynamic*, hence we consider its peak memory per level. Notably, the median memory needs of MINT-LS do not grow significantly with $T$, while the maximum rises as the peak queue size grows with $T$, yet remains under that of SIEVE-Middlepath, the least memory-demanding SIEVE variant. We show results only vs. path length $T$, since subsampling the state space in real data does not significantly affect the performance of MINT-LS, as Figure 3 illustrates.

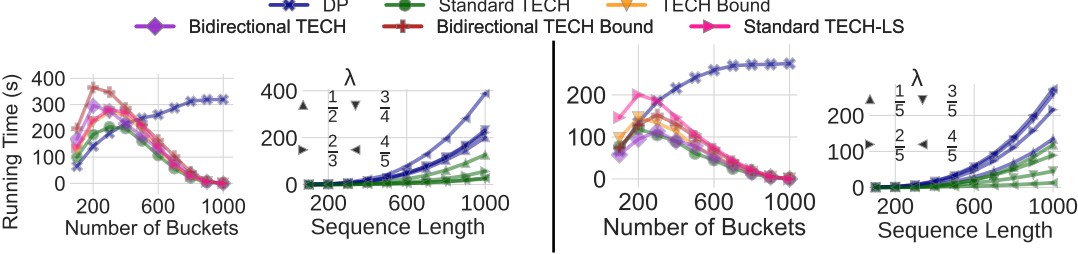

Figure 5: Histogram construction, synthetic data (L), real data (R); runtime vs. $B$ and $n$.

## 4.2 RESULTS ON HISTOGRAM CONSTRUCTION

Figure 5 shows the histogram construction runtime on synthetic data vs. $B$ for input sequence length $n = 1010$, and that of Standard TECH and the standard V-OPT algorithm vs. $n$ for different values of $\lambda = \frac{B}{n}$. Standard TECH often suffices, as its variants do not bring significant advantages. Linear-space TECH incurs a manageable runtime overhead for the sake of space efficiency. Still, all Isabella variants gain in time efficiency as $B$ grows. This growth delimits the search space for each bucket, rendering the problem more amenable to BeFS. Contrariwise, standard DP cannot contain the search space, hence its runtime surges for large $B$. Figure 5 also shows results on real data. The gains of TECH solutions are *more emphatic* here, already expressed with fewer buckets, as TECH exploits data similarity patterns that facilitate summarization, whereas standard DP lacks such capacity.

## 5 CONCLUSION

We introduced Isabella, a framework that efficiently solves problems of fixed-length path optimization by best-first search, while delimiting its space complexity by depth-first search and a divide-and-conquer strategy. We designed an Isabella-based algorithm for Viterbi decoding and one for histogram construction, with bidirectional and bounding variants. Our experiments evince that Isabella gains up to four orders of magnitude in time and space efficiency, the advantage being more pronounced on real data with nonuniform path cost distributions. In Appendix D, we apply Isabella to temporal-graph community search.

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

## A  PROOFS

**Proposition 1.** *Standard MINT is correct.*

*Proof.* Let $\mathcal{T}$ be the set of all tokens (i.e., state-frame pairs) and $\mathbf{V}$ the set of visited tokens, for which optimal cost has been computed. Initially $\mathbf{V}$ includes the source $s$ with optimal cost $p_s = -\log B_{s,y_1}$, and $-p_s$ is its log-likelihood. In each iteration, we add to $\mathbf{V}$ a token $(s_i, t)$ from $\mathcal{T} \setminus \mathbf{V}$ with cost $p_i$. To complete the induction, we must show that $p_i$ is optimal for its length $t$. First, if the true $t$-length-optimal path goes only through tokens in $\mathbf{V}$, for which, by the inductive hypothesis, the optimal cost is known, then $p_i$ must be $t$-length-optimal. Assume that the $t$-length-optimal path goes through a token $(s_j, t')$ not in $\mathbf{V}$; this token necessarily has cost $p_j \geq p_i$, hence any path through it is suboptimal. Therefore, $-p_i$ is the maximum Viterbi path log-likelihood $\max_{s_i} \mathbf{T}[s_i, t]$. $\square$

**Proposition 2.** *MINT Bound is correct.*

*Proof.* We prove the statement by contradiction. Assume the algorithm returns a token $(s^*, T)$ with cost $c^* > \max_{s_i} \mathbf{T}[s_i, T]$. Then there must be an unvisited token $(s', t')$, which, if propagated to the last frame, produces a path of likelihood $\max_{s_i} \mathbf{T}[s_i, T]$. Such a token would have priority $c' + (T - t') \cdot \hat{c}_1$ where $c'$ is the real cost of arriving to $s'$ in $t'$ frames. Since $\hat{c}_1$ lower-bounds the cost of moving from frame to frame, it follows that $c' + (T - t') \cdot \hat{c}_1 \leq \max_{s_i} \mathbf{T}[s_i, T] < c^*$ at any time $t' \leq T$. Therefore $(s^*, T)$ cannot be dequeued before $(s', t')$, ergo the proof is completed. $\square$

**Proposition 3.** *Bidirectional MINT is correct.*

*Proof.* The algorithm alternates between a forward and backward step and maintains the best-so-far path of $T$ steps. Correctness rests on the stopping condition. The algorithm terminates when either (i) both queues are empty or (ii) the elements $(s_i^f, t^f)$ and $(s_i^b, t^b)$ popped from the queues have joint cost $d_f[(s_i^f, t^f)] + d_b[(s_i^b, t^b)]$ that is larger than the current best path cost $\mu$. Regarding condition (i), when both queues are empty, all possible paths have been generated, so the algorithm returns the optimal. Regarding condition (ii), assume that the optimal path is not yet found when the algorithm terminates returning $-\mu$. Then there must be a path $Q^*$ of cost $\mu^* < \mu$ containing at least one not yet visited token in $\mathbf{V}_f \cup \mathbf{V}_b$. Such a token would have cost at least $d_f[(s_i^f, t^f)]$ on the forward side and $d_b[(s_i^b, t^b)]$ on the backward side, hence path $Q^*$ would have cost $\mu^* \geq \mu$, a contradiction. $\square$

**Proposition 4.** *The minimum error* $\min_b E_b$ *among $b$ buckets of an equal-width partitioning a sequence $I$ is a lower bound to the V-optimal histogram cost and $\sum_b E_b$ is an upper bound.*

*Proof.* Let $H_B$ be the V-optimal histogram of size $B$ on sequence $I$ and $H'_B$ be the histogram of the same number of buckets on the same sequence, where all buckets have the same size, except possibly the last. Let $[s_j, e_j]$ be the boundary positions and $E_j$ the error of the $j^{\text{th}}$ bucket in $H_B$, and let $[s'_j, e'_j]$ and $E'_j$ be the corresponding boundary positions and error of the $j^{\text{th}}$ bucket in $H'_B$. Then at least one bucket of $H_B$, say the $j^{\text{th}}$, has $a_j \leq a'_j$ and $b_j \geq b'_j$, i.e., it fully contains the corresponding bucket in $H'_B$. Then, as $E$ is monotonically non-decreasing with bucket width, $E_j \geq E'_j$; besides, $E_j \leq \sum_{h \in H_B} E_h$, ergo $\min_h E'_h \leq E'_j \leq E_j \leq \sum_{h \in H_B} E_h$, hence $\min_h E'_h$ is a lower bound to the V-optimal histogram cost. Furthermore, $\sum_h E'_h$ is an upper bound on the V-OPT histogram cost, since, by definition of the V-OPT histogram $H_B$, $\sum_h E_h \leq \sum_h E'_h$. $\square$

## B  DATA, MEASURES, AND PARAMETERS

**Data.** We experiment on both synthetic and real-world datasets. We evaluate MINT and TECH variants on synthetic data generated according to the following models:

- *Erdős–Rényi* model where each hidden state is emitting and connected with any other state with probability $p = 0.01$; transition and emission probabilities are generated uniformly at random, thus arbitrary cycles may be present. All states are emitting.

- *Skewed path likelihood* model, where we generate a fixed number $N_{path}$ (100, by default) of paths of length $T$ starting from initial state $s$, and composed of emitting states. To each such path we assign a probability drawn from a power law distribution $p(x, \alpha) = \alpha x^{\alpha-1}$, which we distribute evenly across transition and emission probabilities of all states in the path. As in the previous case, all states are emitting. We use this model to investigate how skew in the distribution of path likelihoods affects the advantage granted by Isabella-based solutions.

Further, we assess MINT on real-world speech recognition data:

- *Wall Street Journal (WSJ) corpus* data: we use a real-world composite HMM for speech-text forced alignment, the process of aligning text to audio recordings, which is also tackled by the Viterbi algorithm. The model is built using the HTK software toolkit (Young et al., 2002) and contains 5529 states (including initial states), out of which 3204 are emitting; it was trained on the WSJ corpus (Paul & Baker, 1992) aiming to align speech recordings from the TIMIT corpus (Garofolo et al., 1993).
- *Resource Management (RM) corpus* data: we use a real-world composite HMM for decoding, trained on the RM speech corpus (Price et al., 1993) and built using the Kaldi software toolkit (Povey et al., 2011), The graph comprises 25 333 states (including initial state) and 175 428 edges, out of which 162 255 also carry emission probabilities. We decode subsets of a simple recorded utterance of up to 100 frames.

The observation sequence $Y$ consists of feature vectors of Mel-Cepstrum cepstral coefficients and their derivatives and emission probabilities are given by multivariate Gaussian mixture models.

Similarly, we evaluate TECH in:

- Synthetic *sequences of integers* in the range $[0, 50]$.
- *Dow-Jones Industrial Average (DJIA) closing values* real data.

**Metrics.** We measure runtime in seconds (s) and memory in bytes (B). In all cases, we report the average over 5 repeated runs.

**Parameters.** Regarding decoding in HMMs, in experiments with synthetic data, we vary $T$ in geometric progression of 9 values from 5 to 30 with $K = 7500$; in experiments with real data, we vary $T$ in geometric progression of 9 values from 5 to 100 with $K$ fixed to the size of the original state space. Furthermore, in synthetic data we vary $K$ in a geometric progression of 5 values from 1000 to 16000 with $T = 10$; in real data, we vary $K$ over 5 values in geometric progression from 1000 (forced alignment) or 1900 (decoding) to approximately the size of the original state space, with $T = 30$. To vary $K$ in real-world HMMs, we sample subsets of the original HMM graph via snowball sampling from a start state $source$. To investigate the combined impact of $K$ and $T$, we also vary them simultaneously. We also vary $K \in \{25 \times 10^3, 30 \times 10^3\}$ and $T \in \{25, 30\}$ to monitor how MINT variants behave during runtime in terms of memory usage and the evolution of path likelihood. In the experiment with the skewed path likelihood model, we also vary the power law parameter $\alpha$ controlling skewness in 17 values from $10^{-2}$ to $10^2$ with $N_{path} = 100$, $T = 10$ and $K = 1500$. In the space-efficient variants of MINT-LS, we set the memory budget $\theta$ to 10% of $K \times T$. However, with Erdős–Rényi data, which call for a larger budget as they reflect a worst-case scenario, we set $\theta$ to 90% of $K \times T$.

For the histogram construction experiments, we vary the parameter $B$ from 100 to 1000, while holding $n$ fixed to 1010. We also consider sequences of length $n$ increasing from 100 to 1000 for the different values of $\lambda = \frac{B}{n}$ indicated in the results. With TECH-LS, we set the memory budget to 10% of $n \times B$.

## C ADDITIONAL EXPERIMENTS

**Runtime vs. $\alpha$.** To demonstrate the effect of path likelihood skew using synthetic data, Figure 6 plots runtime as a function of the parameter $\alpha$ of the power law distribution over the path likelihoods. Notably, the highest savings are obtained for $\alpha$ close to 1. This is due to the fact that, for remarkably smaller or larger $\alpha$, all paths tend to be equally likely, so MINT cannot focus on a small subset of paths. Nevertheless, even in the case where the path likelihood distribution approaches the worst case, as in the Erdős–Rényi model, we still have high savings for small $T$, which is a popular

setting in modern speech recognition. Regarding different implementations of MINT, we observe that the use of lower bounds is not always crucial for runtime; however, as $\alpha$ increases, MINT Bound vastly outperforms Standard MINT by virtue of its capacity to prune paths from consideration more aggressively. Furthermore, bidirectional-search variants accomplish the highest efficiency on synthetic data, both those generated by the Erdős–Rényi model and those with skewed path likelihoods; these results vindicate our development of those enhanced solutions.

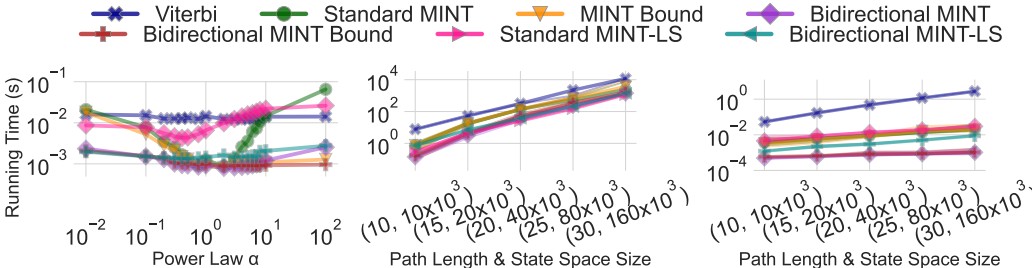

Figure 6: Decoding; synthetic data with skewed path likelihoods; runtime vs. skew $\alpha$ (left); synthetic Erdős–Rényi data (center) and synthetic data with skewed path likelihoods (right): runtime vs. both linearly growing path length $T$ and exponentially growing state space size $K$, indicated as $(T, K)$; shaded regions indicate the min and max runtime due to randomness in data generation.

**Runtime vs. $T$ and $K$ tuned in unison.** We also measure runtime as a function of both $T$ and $K$ on Erdős–Rényi model data and on those with skewed path likelihood distribution with $\alpha = 1$. Figure 6 shows the results. Shaded regions indicate the minimum and the maximum over repeated experiments, which convey the extent of random variation; as the figure shows, that extent is quite limited. The savings observed as we increase both $T$ and $K$ are consistent with our previous findings and most pronounced in the skewed likelihoods scenario. In the Erdős–Rényi model, as most paths of a given length have similar likelihoods, the savings are more modest and decrease with the growth of both $T$ and $K$.

**Real-time memory monitoring.** Figure 7 shows memory requirements at run time for four parameter configurations on synthetic data with skewed path likelihood distribution using $\alpha = 1$; for reference, we also provide the constant memory used by standard Viterbi. Notably, MINT variants reduce the memory requirements of Viterbi by several orders of magnitude. Unsurprisingly, the two bidirectional-search variants consume slightly more memory, yet need fewer iterations till termination, as they apply both a forward and a backward search with two queues. With MINT-LS, we show memory consumption vs. iterations or DFS calls. While by the chosen budget, MINT-LS variants use as little as 25% of the memory used by MINT variants; this advantage may grow on demand by reducing the memory budget $\theta$.

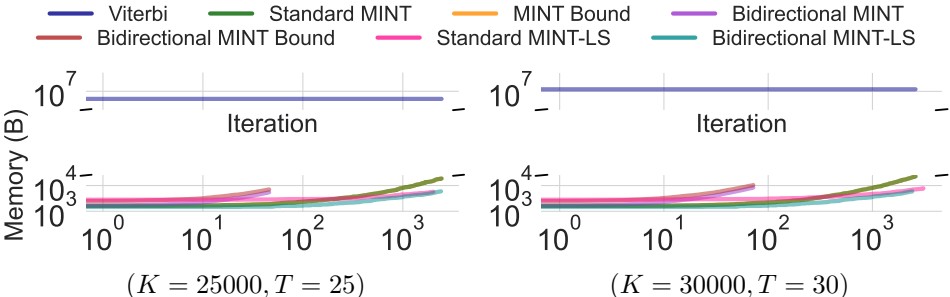

Figure 7: Synthetic data with skewed path likelihood distribution; memory requirements on the fly.

**Effect of backtracking.** As explained in Section 3.2, by default all MINT variants store paths explicitly; however, we may reduce memory usage by only storing the predecessor of each token $(s_i, t)$ and eventually reconstructing the optimal path by backtracking over such links, with a small runtime overhead and savings in memory consumption. We refer to the resulting alternative implementation of MINT as MINT-Backtracking. To illustrate this effect, Figure 8 presents the maximum memory usage of the two implementations of standard MINT under the HMM graph model with skewed path likelihood distribution ($\alpha = 1$) as a function of both $K$ and $T$. While the difference in memory requirement is evident, we measured the corresponding runtime difference to be negligible. Hence,

when memory represents a hard constraint, MINT-Backtracking provides an attractive tradeoff between runtime and memory needs. Besides, MINT-Backtracking extends seamlessly to all variants of MINT. In the case of bidirectional-search-based variants, backtracking proceeds in both directions after the optimal path is found.

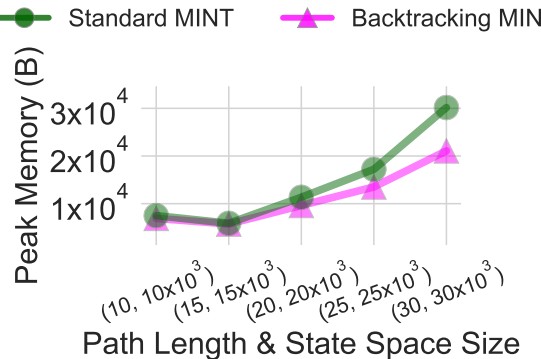

Figure 8: Decoding, synthetic data with skewed path likelihood distribution. Memory requirements of default MINT and more memory-efficient implementation MINT-Backtracking. Maximum (peak) memory vs. path length $T$ and state space size $K$, indicated as $(T, K)$. Ratio $\frac{K}{T}$ is fixed.

**Real-time log-likelihood monitoring.** We also monitor the optimal path log-likelihood absolute value across iterations. This *absolute* value grows, as longer paths have *lower* likelihood than shorter ones. Figure 9 presents our results, using the same four parameter configurations as in Figure 7. In all algorithms the likelihood approaches the optimal value swiftly and monotonically. In the case of standard Viterbi, we plot the highest likelihood found at the end of each frame (i.e., path length considered), hence Viterbi appears to undergo fewer iterations. For the two bidirectional MINT variants, we plot the sum of likelihoods associated with the last tokens de-queued from the forward queue $\mathbf{Q}_f$ and the backward queue $\mathbf{Q}_b$ in each iteration. For other algorithms, we plot the likelihood associated with the token dequed from $\mathbf{Q}$ in each iteration. We found that MINT-LS variants exhibit the same progression of path likelihood as the corresponding MINT variants.

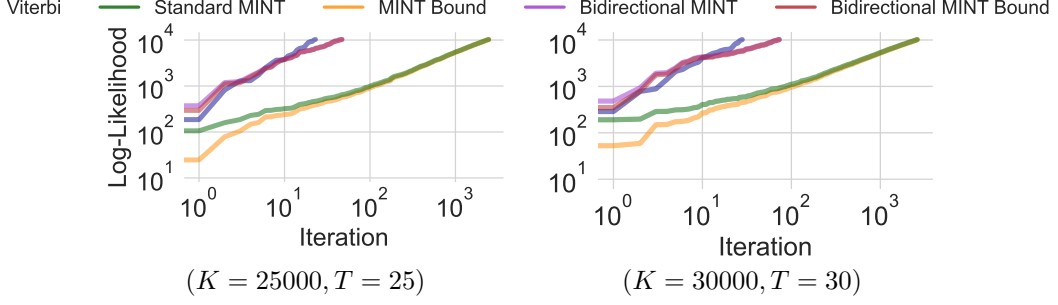

Figure 9: Synthetic data, skewed path likelihood distribution; path log-likelihood (abs) on the fly.

## D    CASE STUDY

We conduct a case study that applies Isabella on an emerging real-world problem of *temporal community search* (Galimberti et al., 2020). Given a temporal graph $G_{\mathcal{T}}$ defined over a temporal domain $\mathcal{T} = [0, 1, \ldots t_{max}]$, an integer $h$, and a set of query nodes $q$, the problem seeks a partitioning $P$ of the temporal domain into $h$ segments and a subgraph $G_h$ containing the query nodes $q$ within each bucket, which yield the maximum *sum of minimum degrees* of subgraphs in $P$. This problem is pertinent nowadays as the growing availability of timestamped data generates interest in temporal graph management. Further, large real-world temporal graphs typically align themselves in evolving communities. To study such communities, it often suffices to focus on a restricted set of query nodes, rather than partitioning the entire temporal graph.

The problem is solved by the dynamic-programming recursion:

$$p^*(i,b) = \max_{0 \le j < t_{max}} p^*(j, b-1) + v_q^*(j+1, i), \tag{5}$$

where $p^*(i,b)$ denotes the optimal objective value for a partition of the first $i$ timestamps in $b$ segments and $v_q^*(j+1, i)$ is the maximum *minimum degree* of a subgraph containing query nodes $q$ and enduring from the $(j+1)^{\text{th}}$ to the $i^{\text{th}}$ timestamp. A cross-examination of Equation 5 and Equation 4 reveals their analogy, with the main difference lying in the value associated with each segment, i.e., in the terminology of Section 3, the *gap function*. Thus, the dynamic-programming algorithm for histogram construction also solves temporal community search with the necessary modifications.

Nevertheless, to compute the gap function $v_q^*(j,i)$ we need to identify a subgraph containing the query nodes $q$ of maximum minimum degree, for each query and each of the $O(t_{max}^2)$ possible $(j,i)$-buckets. The solution in (Galimberti et al., 2020) precomputes all gap function values through *span-core decomposition* and uses them in the dynamic-programming recursion of Equation 5.

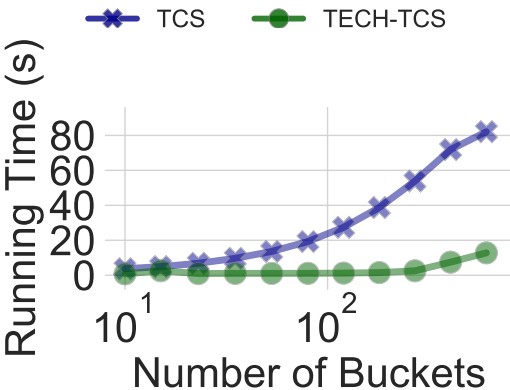

Figure 10: Temporal community search, real-world interaction data; runtime by number of buckets the temporal domain is partitioned into; x-axis on log scale.

We apply Isabella to obtain an advantage over the DP-based solution to temporal community search on a real-world temporal network dataset that captures interactions between students and teachers of nine classes recorded during five days in a high school in France (Mastrandrea et al., 2015). The temporal graph has $47.590$ temporal edges (interactions) and $327$ nodes (students and teachers). These parameters only affect the offline pre-computation of gap function values and not the query processing phase. The length of the sequence to be partitioned is $t_{max} = 1212$. We apply the standard DP algorithm (TCS) and an algorithm based on Standard TECH (TECH-TCS) on the problem with a query comprising the node labelled 1. Figure 10 illustrates our runtime results vs. the number of buckets that partition the temporal domain, varied in geometric progression with ratio 1.5, from 10 to 608. Notably, TECH-TCS outpaces TCS, even for a few buckets. We obtained similar results with different query nodes and larger query node sets, as the query affects the pre-computation stage but not the search space of the DP-based solution and its Isabella-based counterpart.

## E  PSEUDOCODES

Here we collect algorithm pseudocodes for MINT-LS.

**Algorithm 4:** MINT-LS

**Data:** HMM graph $G$, transition and emission probabilities $A$ and $B$, states $S$, observations $Y$, ordered time frames $F$, queue size threshold $\theta$, initial and final state $startSt$ and $lastSt$.

1   $\tau \leftarrow \lceil (F[0] + F[-1])/2 \rceil$, $\mathbf{V} \leftarrow \{\}$ // initialization
2   $\mathbf{Q} \leftarrow Queue((startSt, F[0]), p\,(startSt, F[0]) = d, pred = -1, middle\_pair = (-1, -1));$
3   **while** $\mathbf{Q} \neq \emptyset$ **do**
4      $(s_i, t), p_i, pred, middle\_pair \leftarrow \mathbf{Q}.pop();$
5      $\mathbf{V}.add((s_i, t));$
6      **if** $t = \tau \land middle\_pair = (-1, -1)$ **then**
7         $middle\_pair \leftarrow (pred, s_i);$ // update middle pair
8      **if** $(t = F.last) \land (lastSt = -1 \lor lastSt = s_i)$ // lastSt $= -1$ if not input
9      **then**
10         $s_{m-}, s_{m+} \leftarrow middle\_pair;$ // extract middle pair
11         $N_p \leftarrow \lceil \frac{F.size()}{2} \rceil;$ // number of frames before the middle pair
12         **if** $N_p > 1$ // continue recursion in predecessors
13         **then**
14            $F \leftarrow F[: N_p];$ // update frames
15            $S_p \leftarrow$ FIND-T-HOPPRED$(s_{m-}, N_p);$ // find predecessors of $s_{m-}$
16            MINT-LS$(G, A, B, S_p, Y, F, \theta, startSt, s_{m-});$
17         $N_s \leftarrow F.size() - N_p;$ // number of frames after the middle pair
18         **print** $(s_{m-}, s_{m+});$ // in-order print of middle pairs
19         **if** $N_s > 1$ // continue recursion in successors
20         **then**
21            $F \leftarrow F[-N_s :];$ // update frames
22            $S_s \leftarrow$ FIND-T-HOPSUCC$(s_{m+}, N_s);$ // find successors of $s_{m+}$
23            MINT-LS$(G, A, B, S_s, Y, F, \theta, s_{m+}, lastSt);$
24      **for** $s_j$ in $G[s_i]$ **do**
25         **if** $(s_j, t+1) \notin \mathbf{V} \land s_j \in S$ **then**
26            $d \leftarrow p_i - \log A_{s_i, s_j} - \log B_{s_j, y_{t+1}};$
27            **if** $\mathbf{Q}[(s_j, t+1)] > d \lor (s_j, t+1) \notin \mathbf{Q}$ **then**
28               **if** $\mathbf{Q}.size() > \theta \land (s_j, t+1) \notin \mathbf{Q}$ **then**
29                  DFS$(G, A, B, S, Y, s_j, pred, t+1, d, middle\_pair, \mathbf{Q});$
30               **else**
31                  **if** $(s_j, t+1) \notin \mathbf{Q}$ **then**
                     $\mathbf{Q}.insert((s_j, t+1), p\,(s_j, t+1) = d, pred = s_i, middle\_pair = middle\_pair);$
32                  **else** $\mathbf{Q}.update((s_j, t+1), p\,(s_j, t+1) = d, pred = s_i, middle\_pair = middle\_pair);$

**Algorithm 5:** DFS

**Data:** HMM graph $G$, transition and emission probabilities $A$ and $B$, states $S$, observations $Y$, initial state $s_i$, predecessor $pred$, initial frame $t$, initial path priority $p_i$, $middle\_pair$, queue $\mathbf{Q}$.
**Result:** updated queue $\mathbf{Q}$.

1   **if** $t = \tau$ **then**
2      $middle\_pair \leftarrow (pred, s_i);$ // update middle pair
3   **if** $t < T$ **then**
4      **for** $s_j$ in $G[s_i]$ **do**
5         **if** $s_j \in S$ **then**
6            $d \leftarrow p_i - \log A_{s_i, s_j} - \log B_{s_j, y_{t+1}};$
7            **if** $(s_j, t+1) \in \mathbf{Q}$ **then**
8               **if** $\mathbf{Q}[(s_j, t+1)] > d$ **then**
9                  $\mathbf{Q}.update((s_j, t+1), p\,(s_j, t+1) = d, pred = s_i, middle\_pair = middle\_pair);$
10            **else**
11               $\mathbf{Q} \leftarrow$ DFS$(G, A, B, S, Y, s_j, s_i, t+1, d, middle\_pair, \mathbf{Q});$ // continue DFS
12   **else**
13      **if** $(s_i, t) \notin \mathbf{Q}$ **then** $\mathbf{Q}.insert((s_i, t), p\,(s_i, t) = p_i, pred = pred, middle\_pair = middle\_pair);$
14      **else** $\mathbf{Q}.update((s_i, t), p\,(s_i, t) = p_i, pred = pred, middle\_pair = middle\_pair);$
15   **return** $\mathbf{Q};$

