# OpenReview forum: "Fast and Space-Efficient Fixed-Length Path Optimization"
_ICLR.cc/2025/Conference — Submitted to ICLR 2025_

### Official Review · Reviewer_HujK · 2024-10-27

**Soundness:** 2
**Presentation:** 3
**Contribution:** 3
**Rating:** 5
**Confidence:** 2

**Summary:**

This paper studies algorithms for finding the highest probability chain along a Hidden Markov Model (HMM), a problem known as Viterbi Decoding. The focus is on HMM with fixed length $L$ (the algorithmic approach relies on $L$ being known). Let us denote the number of internal states in each of the $L$ layers by $n$.

The classical approach for solving problems of this type is based on dynamic programming. The idea is to iterate the following for $L$ rounds: for each internal state, compute the highest probability reaching this state as the maximum, over all $n$ states of the previous layer, of the probability of reaching each state times the probability of the edge between them. The computational cost for a single node in a single layer is $O(n)$ (for a total of $O(n^2 L)$ for all nodes in all layers), but the space complexity is $O(1)$ for each node since we need to memorize paths (so the total space is a possibly prohibitive $O(nL)$. There is also a simple solution with this kind of memoization, but its running time is $O(n^2 L^2)$. The question is whether one can do much better, and it was answered to the positive by recent works of Ciaperoni et al. (SIGMOD 2022 & Interspeech 2024). They obtained an algorithm with runtime roughly $O(n^2 L)$ and space $O(n)$.

The current paper obtains another algorithm for Viterbi decoding (and a related problem of histogram construction). To my understanding it has similar worst case guarantees to Ciaperoni et al., but the main "claim to fame" of the paper is better practical applicability due to adaptivity to beyond worst case structure of the specific problem instance. Specifically, the algorithm run bidirectionally, both forward and backward, running a Dijkstra type algorithm from each side plus existing techniques in the literature. The algorithmic idea is that for realistic instances, we expect the subset of close to optimal paths to meet much faster than the progress of most other paths, which effectively prunes away a lot of the slowly proceeding paths.

The authors provide a couple of algorithms based on these insights -- one that is competitive in terms of running time and the other that optimizes on space (guarantees are similar to the SOTA as described in the first paragraph). They show the empirical performance of the algorithm as compared to the vanilla Viterbi algorithm.

**Strengths:**

- A new, practical algorithm for Viterbi decoding, an interesting problem with a variety of applications.

- Convincing experimental results (in the supplementary) against existing algorithms.

**Weaknesses:**

- The paper does not contain a good formal discussion of the running time and space bounds, nor theoretical explanation for the good performance on realistic instances.

- The main experimental results are for synthetic random graphs, and for some reason only compare the new algorithms to vanilla Viterbi, and not, say, to SIEVE variants (Ciaperoni et al.). It does look that the supplementary contains additional interesting experiments - I suggest the authors to restructure the experimental section, and perhaps put more focus on it (and less on the different variants of the algorithm).

**Questions:**

- Can you state clearly the running time and space complexity of each of your variants of the algorithm?

- What kind of theoretical analysis do you think can help explain the claimed good practical performace on realistic instances?

---

> ### Comment · Reviewer_JzWT · 2024-11-13
> **Disagreement on review opinion from another official reviewer.**
>
> First, I agree with the reviewer's rating, and I believe the paper falls below the acceptance threshold. Second, I want to address several disagreements.
>
> - The paper should be evaluated based on the main content. Supplementary can assist us to improve understanding but shouldn't be a factor of strengths.
> - Regarding weakness, I disagree with the reviewer's opinion. I don't think the authors need a formal discussion of runtime and space bounds. First, considering the context, these are fairly trivial, don't warrant a theorem or proposition. Second, the authors briefly discuss complexity in the main paper, which, I believe, is already sufficient. Additionally, the paper includes experiments demonstrating practical improvements in runtime and space, so theoretical analysis is less relevant given they didn't claim contributions on this aspect. The authors care more about the algorithmic design and practical improvements.
> - I also disagree with the reviewer on the dataset critique. In the main paper, they did give discussions on real data, please refer to p.9-p.10.
>
> The authors frequently emphasize the uniqueness of the work is from a novel combination of strategies (bfs, dfs, d&c ...), and provide experiments showing the due improvement. My concerns are: 1) Why is the combination important? Perhaps no one has done the similar before, but the combination doesn't appear particularly technical. 2) What do the reported improvements signify? Are the baselines the latest exact solvers for the two problems? Because, it seems the authors use most of the efforts to make the comparison under their framework with minor variations. (standard MINT, MINT Bound, Bidirectional MINT...)
>
> To be honest, I can only evaluate the technical aspect, specifically, I am confident at the algorithm design aspect and the complexity aspect. I understand their approach. But it doesn't mean I am confident in their application aspect, or if their algorithm design is genuinely significant for fixed-length path optimization problems. I am not familiar with the state-of-the-art exact solvers for Viterbi Decoding or histogram construction. So maybe we should suggest ACs to re-evaluate the experimental results.
>
> Please note that all my comments are intended to improve the review quality. I would be happy to raise my rate if I am proven wrong.

---

> ### Author Response · Authors · 2024-11-15
>
> Thank you for your elaborate review.
>
> >The paper does not contain a good formal discussion of the running time and space bounds
> >Can you state clearly the running time and space complexity of each of your variants of the algorithm?
>
> For a discussion of the running time and space bounds, see Lines 167-184.
>
> > nor theoretical explanation for the good performance on realistic instances.
> >What kind of theoretical analysis do you think can help explain the claimed good practical performace on realistic instances?
>
> The good performance arises due to the nature of real-world problems, as discussed in Lines 469-472, 485-486, 527-529, and 536-538. In real data paths costs follow a skewed distribution, hence the best-first search explores promising paths in higher priority and quickly terminates to the optimal solution.
>
> >The main experimental results are for synthetic random graphs
>
> We do offer more space to results on real graphs (Figures 3 and 4) than on synthetic graphs (Figure 2).
>
> >only compare the new algorithms to vanilla Viterbi, and not, say, to SIEVE variants (Ciaperoni et al.).
>
> We do compare to SIEVE in Figure 4. We omitted a runtime comparison to SIEVE in Figures 2 and 3, given that SIEVE has higher runtime than Vanilla Viterbi, as Figure 4 shows.

---

> > ### Comment · Reviewer_HujK · 2024-11-20
> > **Thank you**
> >
> > Thanks for your response. I still believe that a better theoretical explanation (that faithfully models the "skewness" of data that you mentioned, and takes it into account in the theoretical analysis) would be useful to make the paper substantially stronger theoretically, and not just an application paper that combines a set of techniques. I am keeping my score for now.

---

> > > ### Author Response · Authors · 2024-11-20
> > > **Real suboptimal paths accumulate high cost well before reaching the final length**
> > >
> > > Thank you for your reply.
> > > The crux of the matter is that best-first search only expands subpaths having cost lower than the optimal path.
> > > In synthetic data where transition and emission probabilities are uniformly distributed, there are subpaths having cost lower than the optimal path all the way to the final legth. In real data such a situation rarely arises, as suboptimal paths accumulate high cost well before reaching the final length.

---

> ### Author Response · Authors · 2024-12-01
> **Expressing our sympathy with you**
>
> Dear Reviewer HujK,
>
> We would like to express our sympathy with you over being attacked by another reviewer over the content of your review.
>
> As you may read in the same forum, we have engaged in a long discussion with the reviewer who attacked you, which revealed a lack of awareness of basic mathematical facts on behalf of this reviewer, such as how an exponential function appears on a logarithmic plot. In the course of this discussion, this reviewer repeatedly claimed that our work to lack technical meaningfulness and important references. When asked to clarify their claims with arguments and references on both accounts, they simply refrained from doing so and shifted the focus of the discussion to another trolling attempt.
>
> Overall, we are quite astonished by the lack of professionalism, frivolous attitude, and mathematical incompetence exhibited by this reviewer, which we have not encountered previously in the process of reviewing. We would therefore like to simply express our sympathy with you over being attacked over the content of your review.
>
> Sincerely,
> The authors

---

> > ### Comment · Reviewer_JzWT · 2024-12-01
> >
> > I am glad to see the authors being wholesome. Unlike the above response from the author is definitely a direct attack to me, my response to this review is not a personal attack.
> >
> > In this response, I am actually defending the authors, if they read it carefully.
> >
> > I would be more than happy to exchange opinions with the reviewer.
> >
> > However, why didn't the authors express their sympathy to the reviewer earlier?
> >
> > I am happy to further discuss the simple and basic log-part. I encourage all aspects carefully review the exchanges between the author and me.
> >
> > Thanks all!

---

> > > ### Author Response · Authors · 2024-12-01
> > > **Thank you for sharing**
> > >
> > > Thank you for sharing your belief that to go beyond your prerogative as reviewer and criticize the quality of another review is not a personal attack to the reviewer.

---

> > > > ### Comment · Reviewer_JzWT · 2024-12-01
> > > >
> > > > You are very welcome. Yes, it is NOT an attack.
> > > >
> > > > Please refer me to the corresponding rules or regulations given by ICLR 2025 on  "your prerogative to another review".
> > > >
> > > > Also, it's not a criticism, it's a "disagreement".

---

> > > > > ### Author Response · Authors · 2024-12-01
> > > > > **Please provide the explicit reference**
> > > > >
> > > > > Please cite the part of the ICLR rules and regulations that prescribe it is the prerogative of one reviewer to publicly judge and criticize the quality of another reviewer's review.

---

### Official Review · Reviewer_7Ehd · 2024-10-30

**Soundness:** 2
**Presentation:** 2
**Contribution:** 3
**Rating:** 6
**Confidence:** 3

**Summary:**

This paper proposes time- and space-efficient algorithms for finding a minimum-cost path having predetermined hop length. Their algorithms combine best-first-search to reduce the computation time and depth-first-search to reduce required space. The applications of this problem include the decoding of hidden Markov models (HMM), which is typically solved by Viterbi algorithm that is a famous dynamic programming (DP) algorithm, and V-optimal histogram construction, which is also typically solved by DP. Experiments show that the proposed algorithms are indeed more time- and space-efficient than the ordinal DP algorithms.

**Strengths:**

The proposed algorithms are extensively tested with both synthetic and real datasets with various parameters. These experiments show that the proposed methods is both time- and space-efficient in solving fixed-length path optimization. It is also favorable that the proposed algorithms works extremely well for the real instances since it visits only a small fraction of search states.

**Weaknesses:**

Presentations of this paper have two major issues, which prevent me from understanding the whole algorithm correctly and thus cannot review whether the described algorithm is correct.

The first issue is around the depth-first strategy to achieve linear space complexity. The procedures for achieving linear space complexity is described in the last two paragraphs of Section 3.1 and Section 3.3. While the former only describes the abstract of used ideas (divide-and-conquer and depth-first-search), the latter explains the procedures to achieve linear space complexity. However, I cannot understand this description because some undefined terms arise. For example,
- (l.345) ... and produce all its derivatives on demand via a DFS traversal ... : What is "derivative"? And, what "on demand" means?
- (l.348) ... The paths DFS explores identify and pass on middle pairs as usual ... : What is "middle pairs"? Is it the same with "middle frame"?
- (l.349) By virtue of this DFS operation, ... : What is the "virtue" of DFS?
In addition, there are no step-by-step explanations or pseudocodes for these procedures, that makes the understanding of this part much difficult. Thus, the authors should describe either a step-by-step explanation or pseudocodes for these procedures.

The second issue is that the details of the algorithm are given only for the specific problems such as HMM decoding and histogram construction and not given for the Isabella framework, thus questionable whether Isabella is applicable for the other problems. As far as I understand, these problems share the components (items 1--7) in lines 141--153. Thus, from the name "Isabella framework" I expect that applying these components automatically yields efficient algorithms such as "Standard XXX", "Bidirectional XXX", "XXX Bound", and "XXX-LS". However, this is not the case within the presentation of the paper; "Standard MINT", "Bidirectional MINT", "MINT Bound", and "MINT-LS" are derived, afterwards "Standard TECH" are derived, and then TECH variants are derived by analogy to MINT variants. Thus, I suspect that we cannot derive these variants for the other problems sharing the components (items 1--7). To alleviate this issue, I recommend that, if possible, first the step-by-step explanations or the pseudocodes based on the Isabella framework (assuming the components in lines 141--153) is given first, and then the algorithms for specific algorithm are derived by substituting the components (items 1--7).

Minor comments:
- Firstly I cannot understand what are the triangle marks associated with a fraction in the second and fourth plots are; please note that they are just legends.

**Questions:**

- The above comment is based on my understanding that the common procedure can be described on the Isabella framework like, e.g., "Standard Isabella", "Bidirectional Isabella", "Isabella Bound", and "Isabella-LS". Is this true? If not, please provide why this is called "framework".
- What is the filled area in the right part of Figure 4?

---

> ### Author Response · Authors · 2024-11-14
>
> Thank you for the time invested for your thorough review.
>
> >(l.345) ... and produce all its derivatives on demand via a DFS traversal ... : What is "derivative"? And, what "on demand" means?
>
> The *derivatives* of a token are the tokens generated from it in subsequence time steps. *On demad* means that we generate those tokens upon request, before their turn to be generated by the regular best-first search procedure arrives.
>
> >(l.348) ... The paths DFS explores identify and pass on middle pairs as usual ... : What is "middle pairs"? Is it the same with "middle frame"?
>
> Yes, a *middle pair* is a pair of nodes connected by a *middle edge* at the *middle frame* of the solution path.
>
> >(l.349) By virtue of this DFS operation, ... : What is the "virtue" of DFS?
>
> "By virtue of" is an English expression that means *because or as a result of*.
>
> >describe either a step-by-step explanation or pseudocodes for these procedures.
>
> Depth-first search (DFS) works as usual; at a node $v$, it visits (i.e., produces a token for) a child node of $v$ and, recursively, its descendants; after it has processed this branch, it moves on to the next child of $v$. Each branch terminates when it either reaches the last frame or injects a token into the token set of a state $s_{j'}$ with the same time step as a pre-existing (i.e., already *discovered*) token in that set, as Lines 346-348 describe. As the operation of DFS is classic textbook material, we refrained from offering a pseudocode. A classic pseudocode is as follows:
>
> DFS($G$, $v$)
>     for each child $w$ of $v$:
>         if $w$ not *discovered*
>             DFS(G, $w$)
>
> >the details of the algorithm are ... not given for the Isabella framework
>
> Please note that the details of the algorithm are given for the Isabella framework in Lines 167-184.
>
> >"Standard MINT", "Bidirectional MINT", "MINT Bound", and "MINT-LS" are derived, afterwards "Standard TECH" are derived, and then TECH variants are derived by analogy to MINT variants. Thus, I suspect that we cannot derive these variants for the other problems sharing the components (items 1--7).
>
> We can indeed derive such variants for other problems sharing the components, exactly by analogy to MINT variants. We present specific variants to show the particulars of an application problem.
>
> >I recommend that, if possible, the step-by-step explanations based on the Isabella framework is given first, and then the algorithms for specific algorithm are derived by substituting the components (items 1--7).
>
> This is exactly the presentation order the paper follows. First, it gives the step-by-step explanation based on the Isabella framework, in Lines 167-184, and then derives the algorithms for specific problems by substituting the components in Sections 3.2 and 3.4.
>
> >what are the triangle marks associated with a fraction in the second and fourth plots.
>
> The triangle marks define the markers in the plot; as the text defines in Lines 523-524, the fractions are values of $\lambda = \frac Bn$, i.e., ratios of the number of buckets to sequece length.
>
> >The above comment is based on my understanding that the common procedure can be described on the Isabella framework like, e.g., "Standard Isabella", "Bidirectional Isabella", "Isabella Bound", and "Isabella-LS". Is this true?
>
> Yes, your understanding is correct.
>
> >What is the filled area in the right part of Figure 4?
>
> As the caption of Figure 4 defines, the shaded regions indicate the range of memory consumed over recursive calls.

---

> > ### Comment · Reviewer_7Ehd · 2024-11-18
> >
> > Thank you for providing replies. Regarding the procedures of DFS, I'm sorry for misunderstanding the expression "By virtue of". For the other comments, I admit that the DFS itself is just a textbook material. However, judging from your comment, the depth-first-search strategy performs more than just performing DFS. Since the authors say that achieving linear space complexity is one of the main contribution of this paper, please provide more elaborated descriptions (possibly including pseudocode for this) for both divide-and-conquer and depth-first-search strategies if accepted.
> >
> > Regarding Isabella framework, although the authors claim that the Isabella framework is described in detail in Lines 167-184, I think these descriptions are only for "Standard Isabella" (analogous to Standard MINT and Standard TECH). The descriptions for other variants, namely "Isabella Bound", "Bidirectional Isabella", and "Isabella-LS", are not generally shown and are shown only for MINT. My claim is that these variants should also be described as a general framework, including the detailed explanations for divide-and-conquer and depth-first-search strategies.

---

> ### Author Response · Authors · 2024-11-18
> **We offer specific derivations of bounds for the two problems we study**
>
> >the depth-first-search strategy performs more than just performing DFS.
>
> DFS indeed performs DFS as usual in the space of tokens. We have updated the paper and included in a pseudocode for the divide-and-conquer and DFS procedures in Algorithms 4 and 5, Appendix E, discussed in Lines 351-360 in Section 3.3.
>
> >descriptions for other variants, namely "Isabella Bound", "Bidirectional Isabella", and "Isabella-LS", are not generally shown.
>
> "Bidirectional Isabella" and "Isabella-LS" are indeed described upon their first apperance in the context of MINT, yet there is nothing specific to MINT in that description. The same description applies to other cases; we will include them in the general framework. On the other hand, the description of "Isabella Bound" depends on the particular problem examined, as bounds are problem-aware. We offer specific derivations of bounds for the two problems we study.

---

> > ### Comment · Reviewer_7Ehd · 2024-11-26
> >
> > Thank you for reply about the Isabella framework. I think establishing framework for fixed-length path optimization (defined by items 1--7) is practically important. So, I will keep the initial rating.

---

> > > ### Author Response · Authors · 2024-12-01
> > > **Thank you**
> > >
> > > Thank you for the positive judgement.

---

### Official Review · Reviewer_JzWT · 2024-11-04

**Soundness:** 2
**Presentation:** 2
**Contribution:** 2
**Rating:** 3
**Confidence:** 4

**Summary:**

Summary: The paper works on fixed-length path optimization problems. Specifically, the authors applied the proposed framework to Viterbi decoding and histogram construction problems. They claim contributions as follows: 1) the combination of the BFS (best-first-search) + DFS (depth-first-search) and divide-and-conquer is novel and has never been done previously, 2) experimental results show that their solutions are highly efficient in both time and space compared to the standard dynamic programming (DP) methods like Viterbi and its variants.

**Strengths:**

Pros: The authors picked the classic fixed-length path optimization problem which is highly related to the field of operations research and has an inherently fundamental impact. They work on the time and space improvements to DP, which has significant impacts on AI algorithms.

**Weaknesses:**

Cons: I outline my concerns on the technical aspect, and presentation aspect. I leave the experimental results for  further discussion due to my lack of hands-on experience with the two problems.

- On the technical aspect, the paper expands on solving dynamic programming in a Bellman "fashion". I believe that this topic is very well-studied [Bellman, R.(1952)][4]. In this paper, BFS prioritizes the promising subproblems, DFS maintains a short priority queue, and divide-and-conquer prevents exhaustive tabulating. Maybe this approach has never been done before, but I don't think it's highly technical.

The authors applied the framework by creating multiple algorithms. I argue that, for the BestFS, the authors replace the BreathFS component from the work [Young et al.][1], the bidirectional idea can be traced back to [Pohl][2], the divide-and-conquer application in Viterbi decoding is originated from [Ciaperoni et al.][3]. Therefore, I am a bit skeptical about the innovation of this work, although the authors claim that their approach has improved magnitudes on both time and space.

I can see the improvements from the experiments, but I also believe that combining $\textit{the best}$ of various methods will always end up with certain improvements. The question is, why is the combination important to the field, why is the combination unique and innovative?

- Is there any particular reason to bring up Semirings and Dioids? If so, which aspects of the work do these concepts contribute to? I can see the only usage is in the descriptions of the algorithms. If so, it's very unnecessary to include them as a part of the paper. Dynamic programming could be explained in a few sentences without losing any promotion of the major contributions. On this one, please correct me if I explicitly missed anything.

- For standard MINT, is there any "major" difference with Dijkstra? Except 1) the cost function for path, 2) fixed length. (I understand the scenario is on HMM and we aim to maximize the log prob of Viterbi Path)

- Then, on the presentation aspect, I briefly list out the issues I noticed:
1) MINT -- Is this an abbrev. that has been formally defined? Or, do we just directly use it as the name?
2) The algorithm blocks are not well-written. Maybe consider using algorithmicx/algorithm2e, and try to avoid mixing math expressions with normal text? The format is strange, i.e., in Alg. 3, the if conditions should use some line breaks to trim the extra lengths.
3. Fig.2: y-axis is not showing the full range.
4. Fig.2: the last semicolon, what does it mean by "axes in log scale"? Does it mean that both x and y axes are log scaled? How do the authors apply the log-scale to the axes?
5. The abbrev. "D&C" is only used at its declaration.
6. In the definition of "MINT Bound" (the end of p.5), what does "til" mean? I guess it's the abbrev. of "till", right?
7. Sec. 3.4: "V-optimal(V for variance)" should be explained at the first appearance.


[1]: https://www.researchgate.net/profile/Steve-Young-8/publication/2516613_Token_Passing_a_Simple_Conceptual_Model_for_Connected_Speech_Recognition_Systems/links/555dc15e08ae6f4dcc8c5b65/Token-Passing-a-Simple-Conceptual-Model-for-Connected-Speech-Recognition-Systems.pdf
[2]: https://www.osti.gov/biblio/1453875
[3]: https://dl.acm.org/doi/abs/10.1145/3514221.3526170
[4]: https://www.pnas.org/doi/abs/10.1073/pnas.38.8.716

**Questions:**

Please refer to the weakness section, my questions are well stated.

---

> ### Author Response · Authors · 2024-11-13
>
> Thank you for the time invested in this thorough review.
>
> >for the BestFS, the authors replace the BreathFS component from the work Young et al., the bidirectional idea can be traced back to Pohl, the divide-and-conquer application in Viterbi decoding is originated from Ciaperoni et al.
>
> Yes, these fragments of ideas have existed previously, yet none of the above has applied *depth-first search* to keep the size of the priority queue in check, and none of them has used these ideas **in combination**.
>
> >why is the combination important to the field, unique and innovative?
>
> This combination is *important to the field* because it succeeds to fruitfully apply best-first search to this classic type of problem and retain the memory consumption of the priority queue in check, reaping benefits in time- and space-efficiency; it is *unique* because such a combination has not been tried in previous work; it is *innovative* because it is not obvious a priori.
>
> >Is there any particular reason to bring up Semirings and Dioids?
>
> Yes, we bring up semirings and dioids to properly define the type of problems we study in all generality; without these concepts, we would have to write continuous reminders that "*multiplication could be used in place of addition*" in many equations, and the generality of the framework would have been blurred.
>
> >For standard MINT, is there any "*major*" difference with Dijkstra? Except 1) the cost function for path, 2) fixed length.
>
> Yes, there is a *major* difference from Dijkstra. As we explain in the introduction, a direct application of Dijkstra to find an optimal-cost path of a predetermined length would raise additional space requirements for (i) tabulating the optimal cost per state and step to enable backtracking of the optimal sequence and (ii) maintaining a correspondingly larger priority queue. Standard MINT ameliorates these requirements with (i) a depth-first search strategy that prevents the priority queue from overextending and (ii) a divide-and-conquer provision that omits tabulating all subproblem solutions.
>
> >MINT -- Is this an abbrev. that has been formally defined?
>
> The name stands for Ti*m*e Eff*i*cie*n*t Vi*t*erbi; we apologize for the omission.
>
> >avoid mixing math expressions with normal text?
>
> Thank you for pointing that out. We will eliminate italics from pseudocodes.
>
> >in Alg. 3, the *if* conditions should use some line breaks to trim the extra lengths.
>
> The *if* statements use **less length** than the page provided by the single-column ICLR template, without any extra length.
>
> >Fig.2: y-axis is not showing the full range.
>
> The axis shows the **full range** of values indeed.
>
> >Fig.2: the last semicolon, what does it mean by "axes in log scale"? Does it mean that both x and y axes are log scaled? How do the authors apply the log-scale to the axes?
>
> Yes, the phrase "axes in log scale" means that both axes are in **logarithmic** scale. We bring  the axes to such scale using the appropriate commands in a plot-drawing package.
>
> >In the definition of "MINT Bound" (the end of p.5), what does "til" mean? I guess it's the abbrev. of "till", right?
>
> Yes, "til" is an English preposition, variant of "till" in the dictionary.
>
> >Sec. 3.4: "V-optimal (V for variance)" should be explained at the first appearance.
>
> The explanation is in the parenthesis: these histograms are called "*V-optimal*" because they optimize variance; the letter 'V' stands for '*variance*'.

---

> > ### Comment · Reviewer_JzWT · 2024-11-21
> >
> > Apologize for the delayed response. I sincerely thank the authors for their reply. I will try to make the best use of the remaining time to communicate with the authors more frequently. I provide my reply to both technical and presentation aspects.
> >
> > The authors ground their contributions primarily on the claim that "the combination is unique and can benefit on both time and space". I leave the uniqueness discussion for other reviewers who are tightly following the latest progress of this "path-finding problem". Beyond that, while the benefits on both time and space are acknowledged, my argument is that it is not novel in the use of "DFS to prevent the PQ from being overly extended". Similar ideas have been widely utilized in all aspects of the algorithmic design, especially branch & bond algorithms, and have been widely applied to various problems, including path optimization problems. Thus, while the combination MIGHT be unique, it is not technical. Same concerns also apply to the usages of BFS and D&C. BFS, DFS and D&C are well-established tools in solving path-finding problems, and it is unsurprised to see combining the best of all can make a certain improvement.
> >
> > And it is also obvious that despite the space reduction on PQ, you still need a certain overhead to update the priorities of the elements in PQ via DFS, right? It reminds me the iterative deepening depth-first search [Korf, Richard (1985)][1] as well. I feel the proposed method is somewhat similar to this one. (generally speaking, not similar in all details)
> >
> > Additionally, I believe the D&C part is not explicitly included in the algorithm block: Algorithm 1(standard MINT), right? I have to refer back to the [Ciaperoni et al. (2022)][2] to see how the D&C is applied in their SIEVE.
> >
> > I encourage the authors to persuade me with the *real technical contribution* of this combination. And I disagree that the combination qualifies as technically significant.
> >
> > On the presentations aspect, I defend my perspective on the application of semirings and diodes. It is not proper to use the terminologies here because there is no related theoretical results in this paper. To explain the idea of the proposed framework, especially just the DP component, using well-established, plain DP terminology is more than sufficient to realize the presentation goal. Just like [Ciaperoni et al. (2022)][2].
> >
> > Based on my understanding so far, this work lacks significant contributions to the field. I appreciate the authors' efforts on empirical aspect and I am willing to lean positively toward their claims of "improvements", but, I really did not see any compelling technical innovation in this work.
> >
> > I encourage the authors list out few more state-of-the-art work in this specific direction. If all have been cited in the paper, please let me know as well.
> >
> > I am happy to have further conversations on the technical aspect with the authors, and I am still open to raise my rating.
> >
> > [1]:https://doi.org/10.1016%2F0004-3702%2885%2990084-0
> > [2]:https://dl.acm.org/doi/abs/10.1145/3514221.3526170

---

> ### Author Response · Authors · 2024-11-22
> **Kindly refer to works using DFS to prevent the PQ from being overly extended in path optimization problems**
>
> Thank you for your engagement in discussion and thoughtful response.
>
> >it is not novel in the use of "DFS to prevent the PQ from being overly extended". Similar ideas have been widely utilized in all aspects of the algorithmic design, especially branch & bond algorithms, and have been widely applied to various problems, including path optimization problems.
>
> Please kindly provide the references to those other works using DFS to prevent the PQ from being overly extended in path optimization problems.
>
> >It reminds me the iterative deepening depth-first search Korf, Richard (1985) as well. I feel the proposed method is somewhat similar to this one. (generally speaking, not similar in all details)
>
> In fact, the proposed method does not perform depth-first iterative deepening (DFID) [Korf 1985], and its orientation is quite the opposite: DFID performs depth-first search to depth $i$ in iterations, discarding previously generated nodes
> and increasing $i$ in each iteration. By contrast, our solution performs selective DFS on demand without a depth constraint. To our knowledge, this type of solution has not appeared in previous work.
>
> >To explain the idea of the proposed framework, especially just the DP component, using well-established, plain DP terminology is more than sufficient to realize the presentation goal. Just like Ciaperoni et al. (2022).
>
> Yes, that is definitely sufficient when introducing a solution to a single problem, as in [Ciaperoni et al. 2022]. As our paper introduces a solution applicable to several problems, we found the terminology of semirings and diodes necessary for the sake of generality.
>
> >list out few more state-of-the-art work in this specific direction. If all have been cited in the paper, please let me know as well.
>
> Yes, all works we know of in this specific direction have been cited in the paper.

---

> > ### Comment · Reviewer_JzWT · 2024-11-25
> >
> > Thanks for the prompt reply. It took me sometime to roughly go through the cited related works(besides SIEVE). And I reviewed the updates the authors made in their appendix. I believe my original understanding on the algorithm design is correct.
> >
> > Techniques:
> >
> > Note: I am not questioning whether the exact same idea has been done before. After reviewing the works cited in the paper, I believe that, likely, the combination is, indeed, unique, to the topic. However, my concern lies in whether it is technically meaningful, or can be traced back to someone having the similar impacts.
> >
> > - For Viterbi decoding problem, I think the authors make the space improvement from $O(KT)$ to $O(K)$, where $T$ is generally much smaller than $K$. Is this correct? If so, in general cases, I think the DFS can in general make a certain times improvements on space. But in the worst case, the proposed method shows no significant difference on the space complexity comparing with SIEVE, right?
> > - It appears that the authors did not confirm if they have explicitly included the Divide-and-Conquer component in the Algorithm 1 (standard MINT). In your previous reply, you mentioned $\textit{"Standard MINT ameliorates ... (ii) a divide-and-conquer provision that..."}$. However, I didn't find the related information in your Algorithm 1 pseudocode block or its description. But I can indeed imagine how it works after reading [Ciaperoni et al. (2022)][1]. I think you only use D&C in your MINT-LS, right? So far, the only place I can explicitly see the D&C is from Algorithm 4 and 5 from the updated appendix.
> > - Line.174-175, the authors omit the $n$ from log term in the second big-O notation, "from $O(nL(n+lognL))$ to $O(nL(n + logL))$". Could this simplification occur because, in the best case, Isabella may produce only $\textit{L}$ tokens, each in constant time?
> > - Another pop-up question: I notice that the authors use the tight bound in some places, i.e. line.45, line.342. Would it be safe and straightforward to say it will always be $KT$ in space? Basically, K states and T "layers". Or is this a formal result proven elsewhere? (I believe this is a just a very straightforward result.) $\textbf{This question is tangential, the authors can feel free to skip this one.}$
> >
> >
> > Experiments:
> > - In Fig. 4, the authors use "Bytes" as the unit of measurement, but even $10^8$ bytes, is just around 100 megabytes. How should the practicality of these results be interpreted, given that $10^8$ bytes is not particularly large? Could this because, in general, Viterbi Decoding happens more frequently on embedded devices, and therefore, even small improvements still matter? If so, I am happy with the claimed improvement. But I suggest the authors should mention this in the paper.
> > - Is there any consensus on the datasets for benchmarking this particular topic? If so, do the authors use them in this paper?
> > - Related to the first point in the previous section, maybe $T={25,30}$ is a bit small range? Will the same improvement still applicable to some hundred-size of $T$? Or why there is no need to make $T$ larger?
> > - Do the authors think it might worth an ablation study to validate "the majority of the space reduction is contributed by D&C [Ciaperoni et al. (2022)][1] or by the claimed novelty -- DFS for PQ?" They both work on the reconstruction. Although, we don't have time for this, but I believe this should be a very important one considering this is an experimental paper.
> >
> > Presentation:
> >
> > - Regarding the previously mentioned log-scale issue, in Fig.2, if the axes are both in log scale, how should I understand the x-axis of the left side diagram, 5, 8, 12, 19, 30, so the authors use a unique base, or....? Maybe "axes in log scale." is only for the right side diagram?
> > - On the usage of semirings and dioids, the authors justify their usage via describing the Isabella framework, which provides a more generalized explain for the following two case studies: Viterbi Decoding and the histogram problem. On this one, I maintain my position, and I don't think it's worthy to introduce another concept for one-off usage, especially it's trivial to explain and irrelevant to the major content. Let's skip this one for now, because it's a relatively minor one to our major divergence.
> >
> > I maintain my original rating to the paper. I personally don't believe "hasn't been done before" == "technically strong and novel". I also welcome and encourage related experts to evaluate this paper.
> >
> > [1]:https://dl.acm.org/doi/abs/10.1145/3514221.3526170

---

> > > ### Author Response · Authors · 2024-11-25
> > > **Please let the rating reflect the acknowledged uniqueness**
> > >
> > > Thank you for your thorough response.
> > >
> > > >After reviewing the works cited in the paper, I believe that, likely, the combination is, indeed, unique, to the topic.
> > >
> > > For the sake of fair reviewing, would you let the rating reflect this acknowledgement of uniqueness?
> > >
> > > >my concern lies in whether it is technically meaningful, or can be traced back to someone having the similar impacts.
> > >
> > > Yes, the contribution is technically meaningful, as the experimental results in the paper demonstrate.
> > >
> > > >the proposed method shows no significant difference on the space complexity comparing with SIEVE, right?
> > >
> > > Yes, that is right; we claim and demonstrate an improvement on **time** efficiency over [Ciaperoni et al. 2002].
> > >
> > > >the authors did not confirm if they have explicitly included the Divide-and-Conquer component in the Algorithm 1
> > >
> > > Yes, we present the divide-and-conquer component after Algorithm 1 for the sake of a pedagogical exposition of ideas.
> > >
> > > >Line 174-175, the authors omit the $n$ from log term in the second big-O notation. Could this simplification occur because, in the best case, Isabella may produce only tokens, each in constant time?
> > >
> > > This simplification occurs because $\log nL = \log{n} + \log{L}$, and the $\log{n}$ term is dominated by the $n$ term.
> > >
> > > >Would it be safe and straightforward to say it will always be $KT$ in space?
> > >
> > > Yes, this is straightforward, given that dynamic programming reserves an array of size $KT$ in memory.
> > >
> > > >Viterbi Decoding happens more frequently on embedded devices, and therefore, even small improvements still matter?
> > >
> > > Yes, that is right; as mentioned, our contribution over [Ciaperoni et al. 2002] is in time efficiency.
> > >
> > > >Is there any consensus on the datasets for benchmarking this particular topic? If so, do the authors use them in this paper?
> > >
> > > Yes, we use the same standard benchmarks as in [Ciaperoni et al. 2022].
> > >
> > > >Will the same improvement still applicable to some hundred-size of $T$?
> > >
> > > Yes, the same improvement is applicable with $T$ values up to 760, as Figure 4 shows.
> > >
> > > >the majority of the space reduction is contributed by D&C Ciaperoni et al. (2022) or by the claimed novelty -- DFS for PQ?
> > >
> > > The DFS for PQ serves to **transfer** the space reduction achieved by [Ciaperoni et al. 2002] to the fixed-length path optimization method that we introduce, yielding a solution that is both time- and space-efficient.
> > >
> > > >in Fig.2, if the axes are both in log scale, how should I understand the x-axis of the left side diagram, 5, 8, 12, 19, 30?
> > >
> > > These are the integral values of a geometric progress with base 5 and ratio 1.56.

---

> > > > ### Comment · Reviewer_JzWT · 2024-11-26
> > > >
> > > > Thanks for the prompt response. I appreciate the authors' efforts. First, I'd like to apologize for overlooking certain experimental configurations. The authors indeed mentioned $\textit{the geometric progress}$ in p.14.
> > > >
> > > > Following the past:
> > > > - I don't think the authors answer my question directly on "the majority of the space reduction is contributed by D&C or by the claimed novelty -- DFS for PQ?". I think the reason the algorithm is practically fast stems from the use of BestFS (prune many states), and D&C (reduce tabulations) contributes much as well. But the "DFS for PQ" part would contribute very less. Do we agree on this? And the paper doesn't seem to offer a discussion on this point.
> > > > - Regarding the time complexity, the proposed method has complexity $O(nL(n + logL))$ on Viterbi decoding, right? But this isn't better than the vanilla DP in the worst case, right?
> > > >
> > > > About the experiment results:
> > > > - First, I summarize the experiment results on runtime. Here, I only focus on the gap between the most basic vanilla Viterbi and the proposed method.
> > > > 	- In Fig.2, the authors show 100x improvements in runtime for path length less than 5, the improvement quickly reduces to 0 when path length is around 12.
> > > > 	- In Fig.3(left), the authors show 100x to 1000x improvements on different MINT variants. On forced alignment dataset, it's stable, but on standard decoding dataset, it quickly reduces to less than x5 (from 10s to about 1s) with path length extends to 100.
> > > > 	- In Fig.4, the authors show their most stable improvements, but the improvements are about from 1s(100s) to around 0.01s(1s) with a reducing leading margin.
> > > > - If the authors emphasize its practical time performance gain, using the SIEVE datasets for benchmarking, is not very sounding, since the time efficiency is $\textbf{NOT}$ the priority of SIEVE. As I see, SIEVE positions itself as "a space efficient Viterbi reformulation", offering space efficiency in D&C fashion without increasing time costs. Basically, time efficiency is never its priority. And SIEVE authors admit in the paper in Sec.5 "... reduces space complexity with a negligible runtime overhead ...", which further corroborated by their experimental results, as seen in Fig.6 and Fig.7. These figures indicate that SIEVE and its variants exhibit no clear improvements in runtime when compared to the baseline "Vanilla Viterbi," which I believe refers to the standard Viterbi algorithm.
> > > > Given this, is it safe to say that the baseline used in the authors' paper is somewhat weak? Because, at the first place, NO SIEVE variants show a strong runtime improvement to the vanilla one. The only exception I can guess is that, except SIEVE and the authors' work, no much other existing research recognizing the exact same dataset? If so, should we also consider comparing with MILP solution to Viterbi Decoding, given that the proposed method is an exact solution? Otherwise, if the authors emphasize their runtime practicality, they should compare with works designed exclusively for time-efficient improvements.
> > > > - (minor) Why did the authors exclude SIEVE as one of the baselines in the Fig.2 and 3 experiments? I believe SIEVE will perform similarly like vanilla Viterbi, so there is no need to?
> > > > - In Fig.4, the authors extend the path length from 100 to 760 (as they mention in previous reply) for real forced alignment data. However, it is unclear why the same path length extension was not applied to the "standard decoding" dataset? Based on Fig.3 (Right), the leading margin reduces as the path length reaches 100. What results might emerge if the path length extends to 760 for "standard decoding" dataset? (760 is a bit strange value to me, given the authors use log-scale at many places. But I understand it just happens during experiments.)
> > > > - Line.751-752, does the worst-case imply that $\textit{T}$ is close to $\textit{K}$? Or the probability of the paths are uniformly distribution? To me, if T gets large enough, the likelihoods of all paths converge to a similar level, right? Please enlighten me what's the worst-case to Viterbi decoding?
> > > > - The authors state they use snowball sampling for real data. So, do they input the sampled HMM graphs to the Viterbi baseline? Or do they input the original graphs to Viterbi baseline? (I assume that the authors' Viterbi baseline == Vanilla Viterbi in SIEVE paper.) Also, this snowball sampling is the one in graph sampling, right? For example, the authors use it at source node, and run few rounds of the snowball samplings to get an induced subgraph? Please clarify this.
> > > > - To me, a hard case for Viterbi decoding might arise when T gets close to K. I did not see any experiments where T is more than K/2. In general, T/K is around 1/100 or 1/1000. Does this ratio matter to the practical performance as well? While I understand we have limited time, there is no need to conduct any experiments. But I wish to hear the authors insightful discussions.

---

> > > > > ### Author Response · Authors · 2024-11-30
> > > > >
> > > > > Thank you for the detailed response.
> > > > >
> > > > > >I don't think the authors answer my question directly on "the majority of the space reduction is contributed by D&C or by the claimed novelty -- DFS for PQ?".
> > > > >
> > > > > This question stems from the false assumptions that (i) DFS for PQ adds to the space reduction achieved by D&C and (ii) the two can be treated independently. As we replied, the DFS for PQ serves to transfer, i.e., **enable**, the space reduction achieved in [Ciaperoni et al. 2002] on the optimization method we introduce.
> > > > >
> > > > > >I think the reason the algorithm is practically fast stems from the use of BestFS (prune many states), and D&C (reduce tabulations) contributes much as well.
> > > > >
> > > > > No, D&C does not contribute to time-efficiency. On the contrary, it incurs a small time overhead as it recursively repeats work that was already done, for the sake of space-efficiency.
> > > > >
> > > > > >But the "DFS for PQ" part would contribute very less. Do we agree on this?
> > > > >
> > > > > We can neither agree nor disagree, since the premise that DFS contributes in an additive manner to space reduction is false.
> > > > >
> > > > > >the proposed method has complexity $O(nL (n + \log{L}))$ on Viterbi decoding, right? But this isn't better than the vanilla DP in the worst case, right?
> > > > >
> > > > > It is the same as Vanilla DP in any realistic worst-case scenario, where $L << 2^n$, and better than Vanilla DP in practice.
> > > > >
> > > > > >In Fig.2, the authors show 100x improvements in runtime for path length less than 5, the improvement quickly reduces to 0 when path length is around 12.
> > > > >
> > > > > It reduces to 0 when the path length is around 30 on these synthetic data.
> > > > >
> > > > > >In Fig.4 ... the improvements are about from 1s(100s) to around 0.01s(1s) with a reducing leading margin. ...
> > > > > >Based on Fig.3 (Right), the leading margin reduces as the path length reaches 100.
> > > > >
> > > > > The leading margin is **increasing**; please note the y-axes are logarithmic in both cases.
> > > > >
> > > > > >using the SIEVE datasets for benchmarking, is not very sounding, since the time efficiency is the priority of SIEVE.
> > > > >
> > > > > These datasets are standard speech recognition benchmarks that have nothing to do with the aims of SIEVE.
> > > > >
> > > > > >SIEVE and its variants exhibit no clear improvements in runtime when compared to the baseline "Vanilla Viterbi". ...
> > > > > >Why did the authors exclude SIEVE as one of the baselines in the Fig.2 and 3 experiments?
> > > > >
> > > > > The first question provides the answer to the second: we did so because SIEVE has higher runtime than Vanilla Viterbi, hence does not serve as a proper baseline for time efficiency.
> > > > >
> > > > > >is it safe to say that the baseline used in the authors' paper is somewhat weak?
> > > > >
> > > > > No, that is not safe to say, because we **do not** use SIEVE as a baseline for time efficiency.
> > > > >
> > > > > >should we also consider comparing with MILP solution to Viterbi Decoding?
> > > > >
> > > > > We are not aware of any work applying Mixed-Integer Linear Programming for Viterbi decoding; that would be an overkill, as the problem is solved well by dynamic programming and thus does not call for a more complicated solution.
> > > > >
> > > > > >they should compare with works designed exclusively for time-efficient improvements.
> > > > >
> > > > > Please specify which works you refer to; we are not aware of any work advancing the time efficiency of Viterbi decoding.
> > > > >
> > > > > >What results might emerge if the path length extends to 760 for "standard decoding" dataset?
> > > > >
> > > > > The advantage remains, as in Figure 4; we expand the path length in Figure 4 to present a more exhaustive comparison in this figure that includes both runtime and memory.
> > > > >
> > > > > >Line 751-752, does the worst-case imply that $T$ is close to $K$? Or the probability of the paths are uniformly distribution?
> > > > >
> > > > > The examined parameter $\alpha$ determines, eventually, the distribution of transition and emission probabilities, as explained in Lines 702-707.
> > > > >
> > > > > >To me, if $T$ gets large enough, the likelihoods of all paths converge to a similar level, right?
> > > > >
> > > > > Not necessarily. Path likelihoods may differ arbitrarily even with very large $T$.
> > > > >
> > > > > >Please enlighten me what's the worst-case to Viterbi decoding?
> > > > >
> > > > > The worst case for Viterbi decoding is that all edges have the same transition probability and all {state, observation} pairs the same emission probability; then all paths are equally likely, so there is nothing to choose between them.
> > > > >
> > > > > >The authors state they use snowball sampling for real data. So, do they input the sampled HMM graphs to the Viterbi baseline? Or do they input the original graphs to Viterbi baseline?
> > > > >
> > > > > As explained in Lines 735-736, we use snowball sampling to sample subsets of the original HMM graph, hence vary the number of states $K$.
> > > > >
> > > > > >snowball sampling is the one in graph sampling, right?
> > > > >
> > > > > Yes.
> > > > >
> > > > > >To me, a hard case for Viterbi decoding might arise when T gets close to K.
> > > > >
> > > > > A $T$ close to $K$ is unrealistic in real-world decoding tasks, while a path of such length may not even exist in a directed HMM graph.
> > > > >
> > > > > >In general, T/K is around 1/100 or 1/1000. Does this ratio matter to the practical performance as well?
> > > > >
> > > > > This ratio does not affect performance; it only makes the setting realistic.

---

> > > > > > ### Comment · Reviewer_JzWT · 2024-11-30
> > > > > >
> > > > > > Thanks for the efforts. I give my responses following the order of the authors answer (top to bottom). It would be convenient to use hotkey to locate the quotations.
> > > > > >
> > > > > > ## Additional questions:
> > > > > >
> > > > > > > "It reduces to 0 when the path length is around 30 on these ..."
> > > > > >
> > > > > > In Fig.2 (Left), when the path length is 12, The runtime of "Standard MINT-LS" is very close to vanilla Viterbi's. More accurately, between path lengths of approximately 22/23 and 24/25, all MINT variants perform either worse than or at best equal to the vanilla Viterbi's runtime. When path length is 30, none of the MINT-variants can beat vanilla viterbi.
> > > > > >
> > > > > > > (major) "The leading margin is increasing; please ..."
> > > > > >
> > > > > > No, I disagree with the explanation. The issue might lie in my phrasing. Using *Vanilla Viterbi* as baseline:
> > > > > > In Fig.3 (Right, runtime), the runtime advantage of MINT and its variants is clearly reducing as the path length approaches 100.
> > > > > > In Fig.4 (Left, runtime), the runtime advantage of MINT and its variants is slightly reducing as the path length increases.
> > > > > >
> > > > > > > (major) "....we did so because SIEVE has higher runtime than Vanilla Viterbi, hence does not serve as a proper baseline for time efficiency." AND "The advantage remains, as in Figure 4; we expand the path length in Figure 4 to present a more exhaustive comparison in this figure that includes both runtime and memory."
> > > > > >
> > > > > > No, I disagree. Fig.4 is for "real forced alignment dataset", but my question specifically concerns the Fig.3 (Right, path length), which is the runtime results for the "standard decoding dataset". My question is, if we allow the path length to grow beyond 100, i.e. 150,200,250..., will the runtime advantage on the "standard decoding dataset" maintain? Clearly, Fig.3 shows that the runtime advantage is quickly reducing, therefore, I don't think *on the standard decoding dataset*, the advantage will remain.
> > > > > >
> > > > > > Additionally, in the previous response, the authors noted "... we do not use SIEVE as a baseline for time efficiency ...", I understand that in Fig.4, the authors might want to emphasize memory-related results (Right). But on the LEFT side of Fig.4, the authors did make the comparisons on runtime with SIEVE and its variants. How should I interpret this part?
> > > > > >
> > > > > >
> > > > > > > "A T close to K is unrealistic in real-world decoding tasks, while a path of such ..."
> > > > > >
> > > > > > Thanks for the explain. Yes, T can't get very close to K, but T can be linearly related to K, for example, K/3, K/4, K/5 ..., which represent a reasonably larger ratio than 1/100, 1/1000. Certainly, we don't need to consider the K/2, or even 3K/4 ratio in DAG.
> > > > > >
> > > > > >
> > > > > > ## Done discussion:
> > > > > >
> > > > > > > "..DFS for PQ serves to transfer,.."
> > > > > >
> > > > > > Yes, I agree with the authors, "DFS for PQ" provides space improvement with logarithmic runtime overhead in return.
> > > > > >
> > > > > > > "No, D&C does not contribute to time-efficiency. ..."
> > > > > >
> > > > > > Thanks for the clarification. It aligns with my understanding.
> > > > > >
> > > > > > > "We can neither agree nor disagree ...."
> > > > > >
> > > > > > My fault on the unclear expression. I should've said that I believe "DFS for PQ" has a negative impact to the time complexity, but will reduce the space in practice.
> > > > > >
> > > > > > > "It is the same as Vanilla DP..."
> > > > > >
> > > > > > Yes, I agree.

---

> > > > > > > ### Author Response · Authors · 2024-11-30
> > > > > > > **The apparent reduction is an artefact of the logarithmic axis**
> > > > > > >
> > > > > > > Thank you for the targeted questions. Please find the answers below.
> > > > > > >
> > > > > > > >In Fig.3 (Right, runtime), the runtime advantage of MINT and its variants is clearly reducing as the path length approaches 100. In Fig.4 (Left, runtime), the runtime advantage of MINT and its variants is slightly reducing as the path length increases.
> > > > > > >
> > > > > > > No, such reduction does not occur. The apparent reduction is an artefact of the logarithmic axis. For example, consider the functions $f(x) = x$ and $g(x) = x + 1$. On a linear axis, the difference between them at x appears as $g(x) - f(x) = 1$, a constant function of $x$. However, on a logarithmic axis, the same difference appears as $\log(g(x)) - \log(f(x)) = \log(x + 1) - \log(x) = \log\frac{x+1}{x} = \log{\left(1 + \frac1x\right)}$, a decreasing function of $x$.
> > > > > > >
> > > > > > > >My question is, if we allow the path length to grow beyond 100, i.e. 150,200,250..., will the runtime advantage on the "standard decoding dataset" maintain?
> > > > > > >
> > > > > > > Yes, it remains, as we already said in our previous reply.
> > > > > > >
> > > > > > > >Fig.3 shows that the runtime advantage is quickly reducing
> > > > > > >
> > > > > > > No, Fig. 3 does not show that; that is an artefact of the logarithmic axis.
> > > > > > >
> > > > > > > >I understand that in Fig.4, the authors might want to emphasize memory-related results (Right). But on the LEFT side of Fig.4, the authors did make the comparisons on runtime with SIEVE and its variants. How should I interpret this part?
> > > > > > >
> > > > > > > You should interpret this part as the runtime counterpart of the memory-related results that we want to emphasize, which we provide for the sake of completeness.

---

> > > > > > > > ### Comment · Reviewer_JzWT · 2024-11-30
> > > > > > > >
> > > > > > > > Thanks for the authors' reply. In this response, vanilla Viterbi = $\bf VV$, MINT and its variants = $\bf MINT(s)$, path length = $\bf PL$.
> > > > > > > >
> > > > > > > > ## remain issue
> > > > > > > >
> > > > > > > > First, log-scale diagram is straightforward to digest. Second, the math explain is not adequate. In Fig.3, VV has a smaller slope, all MINT(s) have larger slopes.
> > > > > > > >
> > > > > > > > Using the same explain from the authors in a more rigorous fashion:
> > > > > > > > $g(x)=x+c, f(x)=kx$, where $c$ and $k$ are both constant. Both $k$ ($k>1$ in this example) and $c$ are not fixed. $k$ is increasing, $c$ is decreasing, based on Fig.3.
> > > > > > > > So we have $gap = \log(g(x)) - \log(f(x)) = \log(\frac{x+c}{kx}) = \log(\frac{1}{k}(\frac{x+c}{x}))$. Take 10 as log's base, if $PL > 100$, $k$ is around 10, we have $\log(\frac{1}{k})$ around -1, and $\log(\frac{x+c}{x})$ is close to 0 depends on $c$, then $gap<0$.
> > > > > > > >
> > > > > > > > So, $c$ and $k$ matters.
> > > > > > > >
> > > > > > > > In Fig.3 (Left, runtime), VV is linearly increasing in the log-scaled graph, meaning it is exponentially increasing (at least within this given scale). Similarly, even on this log-scaled diagram, MINT(s) is increasing with a clear larger first-order derivative, meaning the runtime increase much faster than VV.
> > > > > > > >
> > > > > > > > I agree that, in Fig.3 MINT(s) still have advantage, for example:
> > > > > > > >
> > > > > > > > - when path length(PL) = 5
> > > > > > > >   - for VV: runtime = 0.9s (estimated in log-scale)
> > > > > > > >   - for MINT(s): runtime = 0.0007s (estimated average in log-scale)
> > > > > > > >
> > > > > > > > - when PL = 47
> > > > > > > >   - for VV: runtime = 9s (estimated in log-scale)
> > > > > > > >   - for MINT(s): runtime = 0.07s (estimated average in log-scale)
> > > > > > > >
> > > > > > > > - when PL = 100
> > > > > > > >   - for VV: runtime = 10s (estimated in log-scale)
> > > > > > > >   - for MINT(s): runtime = 1s (estimated average in log-scale)
> > > > > > > >
> > > > > > > > What the authors are emphasizing is:
> > > > > > > > - PL=5, VV - MINT = 0.9 - 0.0007 = (around) 0.9
> > > > > > > > - PL=47, VV - MINT = 9 - 0.07 = (around) 8.9
> > > > > > > > - PL=100, VV - MINT = 10 - 1 = 9
> > > > > > > >
> > > > > > > > And yes, the *absolute value* is increasing.
> > > > > > > >
> > > > > > > >
> > > > > > > > I emphasize that, as PL increases:
> > > > > > > > - 5->47 (3 log segments):
> > > > > > > >   - for VV: 9/0.9 = 10x
> > > > > > > >   - for MINT(s): 0.07/0.0007 = 100x
> > > > > > > > - 47->100 (1 log segments):
> > > > > > > >   - for VV: 10/9 = 1.1x
> > > > > > > >   - for MINT(s): 1/0.07 = 14.2x
> > > > > > > >
> > > > > > > > Despite MINT(s) has a certain lead within the range, MINT(s) also slow down faster than VV.
> > > > > > > >
> > > > > > > > Based on both math assumptions and actual data estimations, I believe it's important to learn how MINT(s) perform when $PL>100$, *on standard decoding dataset*. In Fig.2 has proven this could happen on some datasets. My assumption is, MINT(s) can only do well when PL is very very small.
> > > > > > > >
> > > > > > > > ## done discussion
> > > > > > > >
> > > > > > > > > "You should interpret this part as the runtime ...."
> > > > > > > >
> > > > > > > > I'm not very persuaded by this explanation. But I give it a pass. Maybe add this explain as a footnote in the paper, it will make things clearer.
> > > > > > > >
> > > > > > > > > No answer
> > > > > > > >
> > > > > > > > The authors didn't address my first question in my previous reply. And I believe there is no further demand for clarification on that question.

---

> > > > ### Comment · Reviewer_JzWT · 2024-11-26
> > > >
> > > > Note (no need to reply this one): I wish our conversation stays coherent to the methodology. Please avoid misleading titles like
> > > >
> > > > > Please let the rating reflect the acknowledged uniqueness
> > > >
> > > > Partial quotations can't rightly represent my position. In my full response, I give my interpretation on the uniqueness, which I do not find particularly critical to contest in the context of the authors' claims. My standing is always: "uniqueness" $\textbf{!=}$ "technically meaningful". Many algorithms use traversal+PQ as a part of their implementations to provide slight improvements, i.e. simple ones like Dijkstra, Prim's variants ... some refined ones like indexed priority queue, which I believe can fit the usage of "updating token in PQ via DFS".
> > > >
> > > > The exchange is to give the paper an objective $\textbf{evaluation} on whether it's "technically meaningful", rather than the final decision. Most importantly, I don't want to leave any preconceived impressions that might influence the ACs/PCs/others' judgments. Instead, I prefer them to read the full exchange and receive a well-rounded picture.
> > > >
> > > > All respect to the authors efforts, I am open to up the rating anytime before the deadline. But I will raise it if I am persuaded.

---

> > > > > ### Author Response · Authors · 2024-11-26
> > > > > **Please clarify what you see as the purpose of this conversation**
> > > > >
> > > > > Dear Reviewer JzWT,
> > > > >
> > > > > In the first response, you declared that:
> > > > >
> > > > > >I am happy to have further conversations on the technical aspect with the authors, and I am still open to raise my rating.
> > > > >
> > > > > Motivated by this declaration, we engaged in a discussion, in the expectation that satisfactory clarifications on our part will lead to a reevaluation of our submission. This discussion now seems to be one of continuously shifting target. After we address and clarify one concern raised, a new concern appears; most recently this is based on the newly introduced concept of **technical meaningfulness**. Could you please clarify what you see as the purpose of this conversation and what you see as **technically meaningless** in our submission?
> > > > >
> > > > > Thank you in advance.
> > > > >
> > > > > Best wishes,
> > > > > The authors

---

> > > > > > ### Comment · Reviewer_JzWT · 2024-11-26
> > > > > >
> > > > > > Previous answer from the authors:
> > > > > >
> > > > > > > Yes, the contribution is technically meaningful, as the experimental results in the paper demonstrate.
> > > > > >
> > > > > > Therefore, in my latest response, I raise my questions on the experimental results. And I expect to resolve those questions with the authors.
> > > > > >
> > > > > > By "technical meaningful", I refer that we resolve the questions with persuasive answers, which is still an ongoing task. I never stated that the submission has anything "technically meaningless", however, merely avoiding "technical meaninglessness" is not sufficient to warrant an 8. Regarding the "shifting target" you mentioned, I think the discussion is coherent so far. And I believe this is the purpose of a rebuttal.
> > > > > >
> > > > > > Please know: While reviewers are willing to update their scores, they should finish the discussions first, and then give the updates on the scores.
> > > > > >
> > > > > > I hope the authors have read my last response with questions, rather than focusing solely on the note regarding their title. It won't yield benefits for either side. I understand the frustration, I wish to continue the discussion with the authors. At the same time, I respect their decision should they feel it is unnecessary to pursue further rebuttals with me.

---

> > > > > > > ### Author Response · Authors · 2024-11-30
> > > > > > > **To Reviewer JzWT**
> > > > > > >
> > > > > > > Thank you for raising questions on the experimental results. Kindly let us know whether we have already resolved those questions or a pending question remains.

---

> ### Author Response · Authors · 2024-11-26
> **Please specify which works you refer to**
>
> Thank you for the detailed response.
>
> >I don't think the authors answer my question directly on "the majority of the space reduction is contributed by D&C or by the claimed novelty -- DFS for PQ?".
>
> This question stems from the false assumptions that (i) DFS for PQ adds to the space reduction achieved by D&C and (ii) the two can be treated independently. As we replied, the DFS for PQ serves to transfer, i.e., **enable**, the space reduction achieved in [Ciaperoni et al. 2002] on the optimization method we introduce.
>
> >I think the reason the algorithm is practically fast stems from the use of BestFS (prune many states), and D&C (reduce tabulations) contributes much as well.
>
> No, D&C does not contribute to time-efficiency. On the contrary, it incurs a small time overhead as it recursively repeats work that was already done, for the sake of space-efficiency.
>
> >But the "DFS for PQ" part would contribute very less. Do we agree on this?
>
> We can neither agree nor disagree, since the premise that DFS contributes in an additive manner to space reduction is false.
>
> >the proposed method has complexity $O(nL (n + \log{L}))$ on Viterbi decoding, right? But this isn't better than the vanilla DP in the worst case, right?
>
> It is the same as Vanilla DP in any realistic worst-case scenario, where $L << 2^n$, and better than Vanilla DP in practice.
>
> >In Fig.2, the authors show 100x improvements in runtime for path length less than 5, the improvement quickly reduces to 0 when path length is around 12.
>
> It reduces to 0 when the path length is around 30 on these synthetic data.
>
> >In Fig.4 ... the improvements are about from 1s(100s) to around 0.01s(1s) with a reducing leading margin. ...
> >Based on Fig.3 (Right), the leading margin reduces as the path length reaches 100.
>
> The leading margin is **increasing**; please note the y-axes are logarithmic in both cases.
>
> >using the SIEVE datasets for benchmarking, is not very sounding, since the time efficiency is the priority of SIEVE.
>
> These datasets are standard speech recognition benchmarks that have nothing to do with the aims of SIEVE.
>
> >SIEVE and its variants exhibit no clear improvements in runtime when compared to the baseline "Vanilla Viterbi". ...
> >Why did the authors exclude SIEVE as one of the baselines in the Fig.2 and 3 experiments?
>
> The first question provides the answer to the second: we did so because SIEVE has higher runtime than Vanilla Viterbi, hence does not serve as a proper baseline for time efficiency.
>
> >is it safe to say that the baseline used in the authors' paper is somewhat weak?
>
> No, that is not safe to say, because we **do not** use SIEVE as a baseline for time efficiency.
>
> >should we also consider comparing with MILP solution to Viterbi Decoding?
>
> We are not aware of any work applying Mixed-Integer Linear Programming for Viterbi decoding; that would be an overkill, as the problem is solved well by dynamic programming and thus does not call for a more complicated solution.
>
> >they should compare with works designed exclusively for time-efficient improvements.
>
> Please specify which works you refer to; we are not aware of any work advancing the time efficiency of Viterbi decoding.
>
> >What results might emerge if the path length extends to 760 for "standard decoding" dataset?
>
> The advantage remains, as in Figure 4; we expand the path length in Figure 4 to present a more exhaustive comparison in this figure that includes both runtime and memory.
>
> >Line 751-752, does the worst-case imply that $T$ is close to $K$? Or the probability of the paths are uniformly distribution?
>
> The examined parameter $\alpha$ determines, eventually, the distribution of transition and emission probabilities, as explained in Lines 702-707.
>
> >To me, if $T$ gets large enough, the likelihoods of all paths converge to a similar level, right?
>
> Not necessarily. Path likelihoods may differ arbitrarily even with very large $T$.
>
> >Please enlighten me what's the worst-case to Viterbi decoding?
>
> The worst case for Viterbi decoding is that all edges have the same transition probability and all {state, observation} pairs the same emission probability; then all paths are equally likely, so there is nothing to choose between them.
>
> >The authors state they use snowball sampling for real data. So, do they input the sampled HMM graphs to the Viterbi baseline? Or do they input the original graphs to Viterbi baseline?
>
> As explained in Lines 735-736, we use snowball sampling to sample subsets of the original HMM graph, hence vary the number of states $K$.
>
> >snowball sampling is the one in graph sampling, right?
>
> Yes.
>
> >To me, a hard case for Viterbi decoding might arise when T gets close to K.
>
> A $T$ close to $K$ is unrealistic in real-world decoding tasks, while a path of such length may not even exist in a directed HMM graph.
>
> >In general, T/K is around 1/100 or 1/1000. Does this ratio matter to the practical performance as well?
>
> This ratio does not affect performance; it only makes the setting realistic.

---

> ### Author Response · Authors · 2024-11-30
> **This work combines time- and space-efficiency as no previous work does**
>
> Thank you for providing a live example of shifting the target, which we pointed out previously. While the discussion was about whether a gap is being reduced in a figure, as soon as we established that the gap is not reduced in the figure, the target shifts to an argument about slopes. However, the mathematical assumptions underpinning this new argument are false.
>
> >In Fig.3 (Left, runtime), VV is linearly increasing in the log-scaled graph, meaning it is exponentially increasing.
>
> No, a linear growth in log-scaled axes does not imply an exponential growth in real values; it only implies a polynomial growth in real values. A polynomial function $y = x^n$ appears as a linear function $\log_b{y} = \log_b{x^n} = n \log_b{x}$ on logarithmic axes and vice versa, as in the present case. On the other hand, an exponential function $y = α^x$ appears also as an exponential function $\log_b{y} = \log_b{α^x} = \log_b{α} \cdot b^{\log_b{x}}$ on logarithmic axes.
>
> Anyway, we understand that the core concern is that the runtime of our solution may reach the runtime of Viterbi at very large path lengths. However, at such great path lengths, our solution has a remarkable and continuously growing advantage of space efficiency vis-à-vis Viterbi, therefore it is again preferable. Figure 4 illustrates this space efficiency advantage. The title of this work is "Fast and Space-Efficient Fixed-Length Path Optimization" exactly because it combines time-efficiency and space-efficiency as no previous work has done.

---

> ### Comment · Reviewer_JzWT · 2024-12-01
>
> ## On Fig.3
> Regarding the interpretation for Fig.3 (Right), my point is clear. The quoted sentence might lack precision, but it's not central to my argument. Yes, the actual growth can be polynomial; the curve appears "exponential", likely because the diagram only shows the initial phase of the complete curves. To put it simply: the trends of $x^n$ and $n^x$ can appear similar, especially when $x$ is small. And that's why I'd like to see $PL>100$.
>
> $\textbf{My core argument is: MINT(s) runtime could surpass that of VV}$ when $PL>100$. I believe my previous math assumptions are sound.
> At this point, on this particular problem, I think things are clear enough. I defend my judgement. And we can wait for the ACs' assessment.
>
> ## On the comparison in experiment sectiob
> In the previous exchange, the authors mentioned: "...we claim and demonstrate an improvement on time efficiency over...".
> In the latest response, the authors mentioned: "...our solution has a remarkable and continuously growing advantage of space efficiency vis-à-vis Viterbi...".
>
> In my opinion, the comparison in experiment section still has much space to improve. Without the rebuttal, it would be hard to discern when the authors prioritize "time advantages" and when they focus on "space advantages."
>
> > "...Figure 4 illustrates this space efficiency advantage..."
>
> I have no issues with the space part in Fig.4. My concerns are on Fig.2 and Fig.3. These are completely different experiments, and I have expressed very different concerns.
>
> ## Conclusion
> Thanks for the authors invested time and efforts during the rebuttal.
>
> Regarding the accusation of "shifting target", I firmly believe my questions are coherent, concise and clear. And I fully refuse the accusation. The rebuttal is to raise, discuss and address the concerns of the paper. Any discussion unrelated to the paper's content is a waste of time, including this paragraph in this response.
>
> Although I hold my opinions about the accusations, including "shifting target" and etc, I have refrained from including those irrelevant arguments in my rebuttal. I suggest the authors set aside judgements about the manner of how I rebuttal, as such remarks do not contribute to the evaluations on this paper.
>
> If the authors believe my review lacks professionalism, they are welcome to escalate this matter to the PCs, ACs and SACs. I am eager to hear ACs' evaluations of this paper, and see if they approve the "shifting target" blame. I have no further questions for now, and stand by my initial assessment. And I will respect ACs' decision.

---

> > ### Author Response · Authors · 2024-12-01
> > **No contradiction between a time-efficiency advantage over SIEVE and a space-efficiency advantage over Viterbi**
> >
> > Thank you for kindly sharing your mathematical beliefs.
> >
> > >$x^n$ and $n^x$ can appear similar on a log-scale, especially when $x$ is small.
> >
> > Unfortunately, this mathematical belief is false; $x^n$ and $n^x$ never appear similar on a log-scale, regardless of the value of $x$. As already explained in our previous reply, $x^n$ appears as a **linear** function on logarithmic scale, while $n^x$ appears as an **exponential** function on logarithmic scale.
> >
> > >In the previous exchange, the authors mentioned: "...we claim and demonstrate an improvement on time efficiency over...".
> >
> > Yes, we claim and demonstrate an improvement on time efficiency over [Ciaperoni et al. 2002]; that is correct.
> >
> > >In the latest response, the authors mentioned: "...our solution has a remarkable and continuously growing advantage of space efficiency vis-à-vis Viterbi...".
> >
> > Yes, that is also correct. There is no contradiction between these two claims. Both of them are true and consistent with each other: we achieve improved time efficiency over SIEVE [Ciaperoni et al. 2002] and also improved space-efficiency over Viterbi, as our experiments establish.

---

> > > ### Comment · Reviewer_JzWT · 2024-12-01
> > >
> > > It is an false deny. $x^n$ and $n^x$ can have similar **trend** in log-scale. Here is the proof:
> > >
> > > https://www.desmos.com/calculator/urtkah5rrt
> > >
> > > The example can not fully reflect the Fig.3 case, but it can expressed my concerns on Fig.3.

---

> ### Comment · Reviewer_JzWT · 2024-12-01
>
> I provide another closer example to the authors to better deliver what I mean. They are not use to fully replicate Fig.3, but can be used to express the overall concerns.
>
> https://www.desmos.com/calculator/qwdki60vve

---

> ### Author Response · Authors · 2024-12-01
> **Thank you for sharing**
>
> Thank you for sharing your belief that a straight line can have a similar trend as an exponential curve.

---

> > ### Comment · Reviewer_JzWT · 2024-12-01
> >
> > In case the link might not work permanently. I post my examples for all aspects to review:
> >
> > In the first link: https://www.desmos.com/calculator/urtkah5rrt
> >
> > - $y=10^{x}$
> > - $y=x^{10}$
> > - $0.00001<x\ <100000$
> >
> > In the second link: https://www.desmos.com/calculator/qwdki60vve
> >
> > - $y=10^{x}$
> > - $y=x^{10} + 100$
> > - $0.00001<x\ <100000$
> >
> > Please feel free to add more polynomial terms, it will better explain my concerns to the Fig.3.
> >
> > Please also refer to our previous exchanges on Fig.3. For the record, the authors use a different log-scale on x-axis on Fig.3, comparing with my above examples.
> >
> > Further response is meaningless, with the direct and clear examples given, I already ask for the help from ACs, SACs and PCs to carefully review the work, and your comments.
> >
> > Despite all the chaos, I still wish the authors can achieve their desired results.

---

> > > ### Author Response · Authors · 2024-12-01
> > > **Please explain how these plots justify your mathematical beliefs**
> > >
> > > Please explain how these plots justify the belief that a function linearly increasing in the log-log plot is exponentially increasing in the lin-lin plot, as expressed in this quotation:
> > >
> > > >VV is linearly increasing in the log-scaled graph, meaning it is exponentially increasing

---

### Official Review · Reviewer_WqbX · 2024-11-09

**Soundness:** 3
**Presentation:** 3
**Contribution:** 3
**Rating:** 6
**Confidence:** 4

**Summary:**

The authors consider the problem of finding a path of a predetermined length that minimizes a cost function in a search space. Traditionally, dynamic programming is used to solve such problems that calls for a lot of time and memory. The authors of this paper present more time and space efficient algorithms.

**Strengths:**

The problem considered in this paper has applications in many domains including bioinformatics, speech recognition, and communication systems. The authors propose a novel framework called Isabella to solve this problem. Isabella combines best-first search, depth-first search, and divide-and-conquer. Each of these techniques is very popular and the idea of combining all of these in the same framework is somewhat interesting. Also, the experimental results reveal that the proposed approach is effective.

**Weaknesses:**

The novelty of the proposed work is modest to some extent.

**Questions:**

None

---

> ### Comment · Reviewer_JzWT · 2024-11-12
> **Disagreement on review opinion from another official reviewer.**
>
> (Redacted by PCs)
>
> Also, I would like to take this chance and have a professional discussion with the reviewer regarding the stated review opinions.
>
> My first question is: what do you see as the novelty of the proposed method, given that you rated both Soundness and Contribution as 3?

---

> ### Author Response · Authors · 2024-11-13
> **Novel combination of classic components**
>
> Thank you for the concise review. The novelty of this work lies in fruitfully applying best-first search (BestFS) to a classic type of problem and devising additional algorithmic contributions to retain low space consumption while maintaining a BestFS heap. The components we use are in themselves classic, yet their combination to achieve the results we present is novel.

---

### Author Response · Authors · 2024-11-22
**Pseudocodes provided as requested**

Dear reviewers,

We have updated the paper with pseudocodes, as requested, especially by Reviewer 7Ehd. Algorithms 4 and 5 in Appendix E now present the space-efficient solution MINT-LS and the DFS subroutine it uses, discussed in Lines 351-360 in Section 3.3.

We hope these pseudocodes address the related concerns.

---

### Comment · Area_Chair_Dd3C · 2024-11-26
**Response**

Dear Reviewers,

The authors have provided their rebuttal to your questions/comments. It will be very helpful if you can take a look at their responses and provide any further comments/updated review, if you have not already done so.

Thanks!

---

### Comment · Program_Chairs · 2024-12-01

The program committee has reviewed the discussion on this forum. We remind all participants to review and follow the code of conduct for the conference and professional standard. Further exchange breaching the norm of the professionalism will cause participants to be removed from the conference. All messages must focus on intellectual discussions and remain professional and polite. Personal attacks and insults are not tolerated at ICLR.

Several comments on this forum have been redacted with visibility only by the SAC and PCs.

---

### Meta-Review · Area_Chair_Dd3C · 2024-12-20

**Metareview:**

This paper propose an algorithm to find a fixed-length path in a network that minimizes a predetermined cost.  While this can be an important contribution, a good theoretical analysis of time ad space complexities are sorely lacking, which contributes to a lack of clarity around the impact of the work.

Based on the reviews and discussions, I recommend rejection.

**Additional Comments On Reviewer Discussion:**

The discussion between authors and a reviewers lacked professional decency. I believe the authors got carried away with their eagerness to see this paper accepted. It was not fruitful.

---

### Decision · Program_Chairs · 2025-01-22

Reject